# The MinDE system is a generic spatial cue for membrane protein distribution in vitro

Beatrice Ramm [1], Philipp Glock [1], Jonas Mücksch [1], Philipp Blumhardt [1], Daniela A. García-Soriano [1], Michael Heymann [1] & Petra Schwille[1]

The *E. coli* MinCDE system has become a paradigmatic reaction–diffusion system in biology. The membrane-bound ATPase MinD and ATPase-activating protein MinE oscillate between the cell poles followed by MinC, thus positioning the main division protein FtsZ at midcell. Here we report that these energy-consuming MinDE oscillations may play a role beyond constraining MinC/FtsZ localization. Using an in vitro reconstitution assay, we show that MinDE self-organization can spatially regulate a variety of functionally completely unrelated membrane proteins into patterns and gradients. By concentration waves sweeping over the membrane, they induce a direct net transport of tightly membrane-attached molecules. That the MinDE system can spatiotemporally control a much larger set of proteins than previously known, may constitute a MinC-independent pathway to division site selection and chromosome segregation. Moreover, the here described phenomenon of active transport through a traveling diffusion barrier may point to a general mechanism of spatiotemporal regulation in cells.

[1] Max Planck Institute of Biochemistry, Am Klopferspitz 18, 82152 Martinsried, Germany. Correspondence and requests for materials should be addressed to P.S. (email: schwille@biochem.mpg.de)

Free energy-driven spatiotemporal organization is key to transforming a pool of molecules into a functional cell capable of exercising complex tasks characteristic of life, such as metabolism and self-replication.

The establishment of spatiotemporal cellular patterns and structures in higher organisms is predominantly mediated through active mechanisms that involve cytoskeletal filaments and motor proteins. Bacteria with their small size and lack of organelle substructures, however, largely rely on reaction–diffusion to orchestrate molecular transport and positioning[1,2]. In particular the MinD/ParA ATPase family is essential for plasmid and chromosome segregation[3], the positioning of FtsZ[4,5] and other protein complexes[6]. The most prominent representative of this protein family is the *Escherichia coli* MinCDE system, which has become a model reaction–diffusion system in biology, extensively studied in vivo[7,8], in vitro[9–11], and in silico[9,12,13]. The MinCDE proteins oscillate from pole-to-pole within the rod-shaped bacterial cell, positioning FtsZ, the scaffold protein for cell division, at midcell[7,14,15]. The ATPase MinD dimerizes upon ATP binding, which enhances its affinity via a C-terminal membrane-targeting sequence (MTS) for the spatial reaction matrix, the membrane[16]. Membrane-bound MinD recruits MinE, which in turn stimulates the ATPase activity of MinD causing MinDE membrane detachment[17]. MinC is not needed for pattern formation, but merely follows the MinDE oscillations[9,14,18]. Thereby, a steady-state concentration gradient of MinC is established with a concentration minimum at midcell[12]. Since MinC inhibits FtsZ polymerization, its spatiotemporal patterning restricts FtsZ ring formation to midcell[19–21].

The oscillatory mechanism for positioning FtsZ by the MinCDE system in *E. coli* is not conserved across prokaryotes. For instance, *Bacillus subtilis* uses a static, polarly localized MinCD system[22]. So why does *E. coli* employ such an eccentric and energy-consuming mechanism? And could the MinDE oscillations have additional roles apart from positioning MinC[23–25]? Several studies reported that MinCDE deletion leads to chromosome segregation defects that cannot be explained by impaired division only[26–28]. In fact, *E. coli* lacks any ParABS system that other bacteria employ for active chromosome segregation, and how exactly *E. coli* segregates its chromosomes is highly debated[24,29,30]. MinD is the closest homolog to ParA in *E. coli* and thus has been suggested to act as driving force for chromosome segregation by direct DNA binding[24]. Another hint for additional roles of the MinCDE system came from the analysis of the *E. coli* inner membrane proteome in Δ*minCDE* and wildtype strains that showed that the abundance of peripheral membrane proteins is regulated by MinCDE[25]. Interestingly, these studies mostly implicate MinDE oscillations, but not MinC, as contributing factors.

Despite these cues, further experimental proof for the extent as well as the underlying mechanism of how MinDE mediate these processes is still lacking. Since MinCDE deletions or manipulations in vivo immediately affect cell division, an unbiased, differentiated functional analysis is nearly impossible. We have therefore turned to an in vitro approach reconstituting MinDE oscillations on supported lipid bilayers (SLBs)[9], where the proteins form traveling surface waves, and in rod-shaped microcompartments[10], where the proteins perform pole-to-pole oscillations mimicking their behavior in vivo. Reducing the system to its core components, MinDE, ATP and the membrane, we directly address MinDE function without the side-effects of component deletion or modification obtained in vivo.

Here, by reconstituting MinDE oscillations in vitro, we demonstrate that their ability of redistributing membrane-attached proteins into steady-state gradients is not limited to direct interaction partners of MinDE. Rather, ATP-driven MinDE self-organization may constitute a dynamic diffusion barrier, causing directed transport of functionally completely unrelated lipid-anchored proteins. Our results imply a much more fundamental role of MinDE in division site selection and chromosome segregation in *E. coli* than simply establishing a MinC gradient and provide the framework for positioning of molecules in artificial cells. Furthermore, our study poses the question whether related reaction–diffusion systems, such as ParABS systems[1], Cdc42[31] and PAR[32] proteins, are also capable of regulating a large set of proteins by similar nonspecific interactions. This may point to a so far unknown generic mechanism of coupling large-scale molecular rearrangements and gradient formation to ATP consumption.

## Results

**MinDE regulate a model peripheral membrane protein**. To test the hypothesis that MinDE oscillations are involved in spatiotemporal positioning of chromosomes and membrane proteins, we used our well-established in vitro reconstitution assay on large planar SLBs, where MinDE form traveling surface waves[9]. We first evaluated the simplest scenario: regulation of monomeric, peripheral membrane proteins by MinDE. We designed a model peripheral membrane protein, mCh-MTS(BsD), consisting of the monomeric, fluorescent protein mCherry[33] and a C-terminal amphipathic helix, the MTS from *B. subtilis* MinD. This MTS is well-characterized and localizes other fluorescent proteins to the inner membrane in *E. coli*, but is unlikely to specifically interact with MinDE[16,34]. When we added this protein to negatively charged SLBs, we observed homogenous membrane coverage (Fig. 1a). Intriguingly, mCh-MTS(BsD) also formed traveling surface waves when co-reconstituted with MinDE. These waves were perfectly anticorrelated with the traveling MinDE waves (Fig. 1a, b, Supplementary Movie 1). When the fluorescence intensity of mCh-MTS(BsD) on the membrane is compared in the presence and in the absence of MinDE, intensities are lower in the presence of the MinDE waves (Fig. 1c). To quantify this effect, we analyzed the mean fluorescence intensity of mCh-MTS(BsD) images for three regions: the full image, and the pixels located in the minima and maxima of the MinDE wave (Fig. 1d, Methods section). Indeed, MinDE waves reduced the overall membrane density of mCh-MTS(BsD), and in particular in the wave maxima (Fig. 1e). Importantly, this spatial regulation of mCh-MTS(BsD) is unlikely to be caused by specific interactions with MinDE (unlike the spatiotemporal regulation of FtsZ filaments by MinDE waves that include MinC[20,35]) and can thus be considered generic.

**Regulation of peripheral membrane proteins is robust**. To demonstrate that the spatiotemporal regulation of peripheral membrane proteins by MinDE is generic, we designed a set of mCherry model membrane proteins (mCh-MTS) with amphipathic helices from different proteins endogenous in *E. coli*: MreB, FtsA and FtsY (MTS(1×MreB)-mCh, MTS(2×MreB)-mCh[36], mCh-MTS(FtsA)[37], MTS(FtsY)-mCh[38]) (Fig. 2a). All mCh-MTS constructs bound to the membrane (Supplementary Fig. 1a, b) and were susceptible to spatial regulation by the MinDE wave, resulting in an anticorrelated mCherry wave on the membrane (Fig. 2b, Supplementary Movie 2). In contrast, the control containing His-mCh, unable to bind to the membrane (Supplementary Fig. 1a, b), showed no spatiotemporal regulation. MTS(1×MreB)-mCh also weakly bound to the membrane and was regulated by MinDE, although in the past no membrane binding of a similar construct could be detected in vivo[36]. To exclude photoinduced artefacts we imaged MTS(2×MreB)-mCh

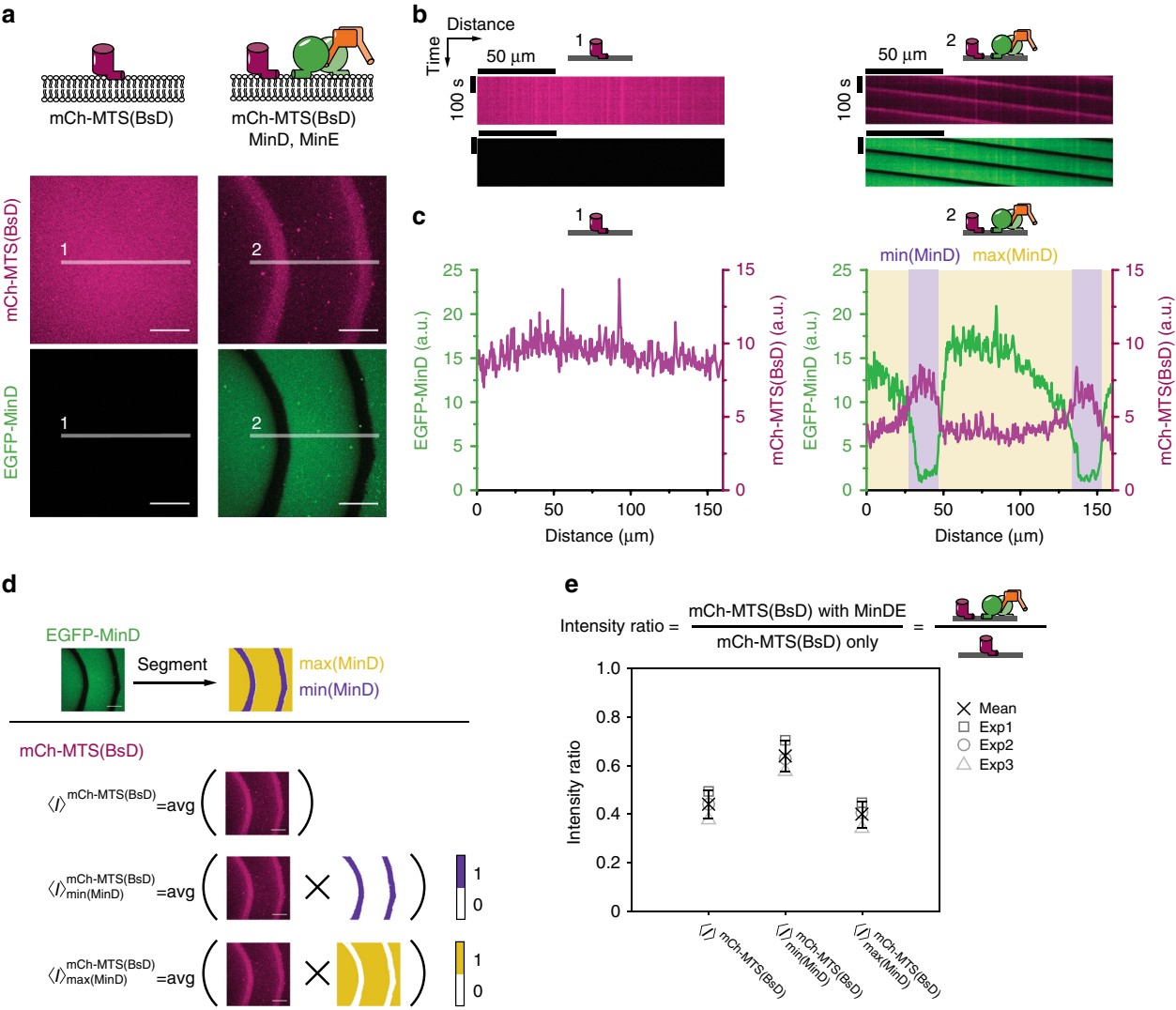

**Fig. 1** MinDE can spatiotemporally regulate a model peripheral membrane protein. **a** mCh-MTS(BsD), mCherry fusion to the C-terminal amphipathic helix of *B. subtilis* MinD, homogenously covers SLBs in the absence of MinDE (1 μM mCh-MTS(BsD)). In the presence of MinDE and ATP mCh-MTS(BsD) forms traveling surface waves that are anticorrelated to the MinDE wave (1 μM mCh-MTS(BsD), 1 μM MinD (30% EGFP-MinD), 1 μM MinE). Scale bars: 50 μm. **b** Kymographs of the line selections shown in **a**. Scale bars: 50 μm and 100 s. **c** Intensity profiles of the line selections shown in **a**. mCh-MTS(BsD) fluorescence (magenta) on the SLBs in the presence of MinDE is reduced and shows clear maxima in the minima of the MinDE waves (min(MinD)) and clear minima in the MinDE wave maxima (max(MinD)). **d** Schematic of the analysis process. EGFP-MinD images are segmented to generate two binary masks that are subsequently multiplied with mCh-MTS(BsD) images to obtain average intensities for the full image and in the minimum and maximum of the MinDE wave. **e** Intensity ratio of the average fluorescence of mCh-MTS(BsD) in the presence over in the absence of MinDE. Intensity ratios are shown for the average intensity of the full image ($I^{\text{mCh−MTS(BsD)}}$), in the MinDE minimum ($I^{\text{mCh−MTS(BsD)}}_{\text{min(MinD)}}$) and in the MinDE maximum ($I^{\text{mCh−MTS(BsD)}}_{\text{max(MinD)}}$). Each data point (exp 1–3) is generated from at least one time series consisting of 75 images in one sample chamber. Cross and error bars depict the mean values and standard deviations from three independent experiments

with non-labeled MinDE and observed the same traveling surface waves for MTS(2×MreB)-mCh (Supplementary Fig. 2a).

The membrane affinity and the extent of spatial regulation differed quite drastically between mCh-MTS constructs (Fig. 2b). To quantify this effect, we analyzed the mCherry and EGFP-MinD fluorescence intensity for the whole image, as well as in the minima and maxima of the MinDE wave (Supplementary Fig. 3) as described (Fig. 1d). We furthermore determined the contrast of the resulting mCh-MTS waves, defined as the average signal in the mCherry maximum above the background ($I^{\text{mCh−MTS}}_{\text{min(MinD)}} − I^{\text{mCh−MTS}}_{\text{max(MinD)}}$) divided by the background intensity ($I^{\text{mCh−MTS}}_{\text{max(MinD)}}$) (Fig. 2c). We assume that the background intensity $I^{\text{mCh−MTS}}_{\text{max(MinD)}}$ is a measure for the overall binding strength

of the mCh-MTS binders (Fig. 2d). $I^{\text{mCh−MTS}}_{\text{max(MinD)}}$ increases from (MTS(1×MreB)-mCh) to (MTS(2×MreB)-mCh), predicted to have the weakest and strongest membrane affinity, respectively (Fig. 2d). Interestingly, the contrast did not directly depend on the binding strength, but the constructs with the highest contrast, mCh-MTS(FtsA) and mCh-MTS(BsD), displayed intermediate background intensity. These two constructs contain a C-terminal MTS like MinD, whereas all other constructs contain an N-terminal MTS. The two termini of mCherry might differ in their flexibility, changing the properties of the mCh-MTS constructs.

Contrary to MinDE pattern formation regulating mCh-MTS constructs on the membrane, MinDE patterns themselves were

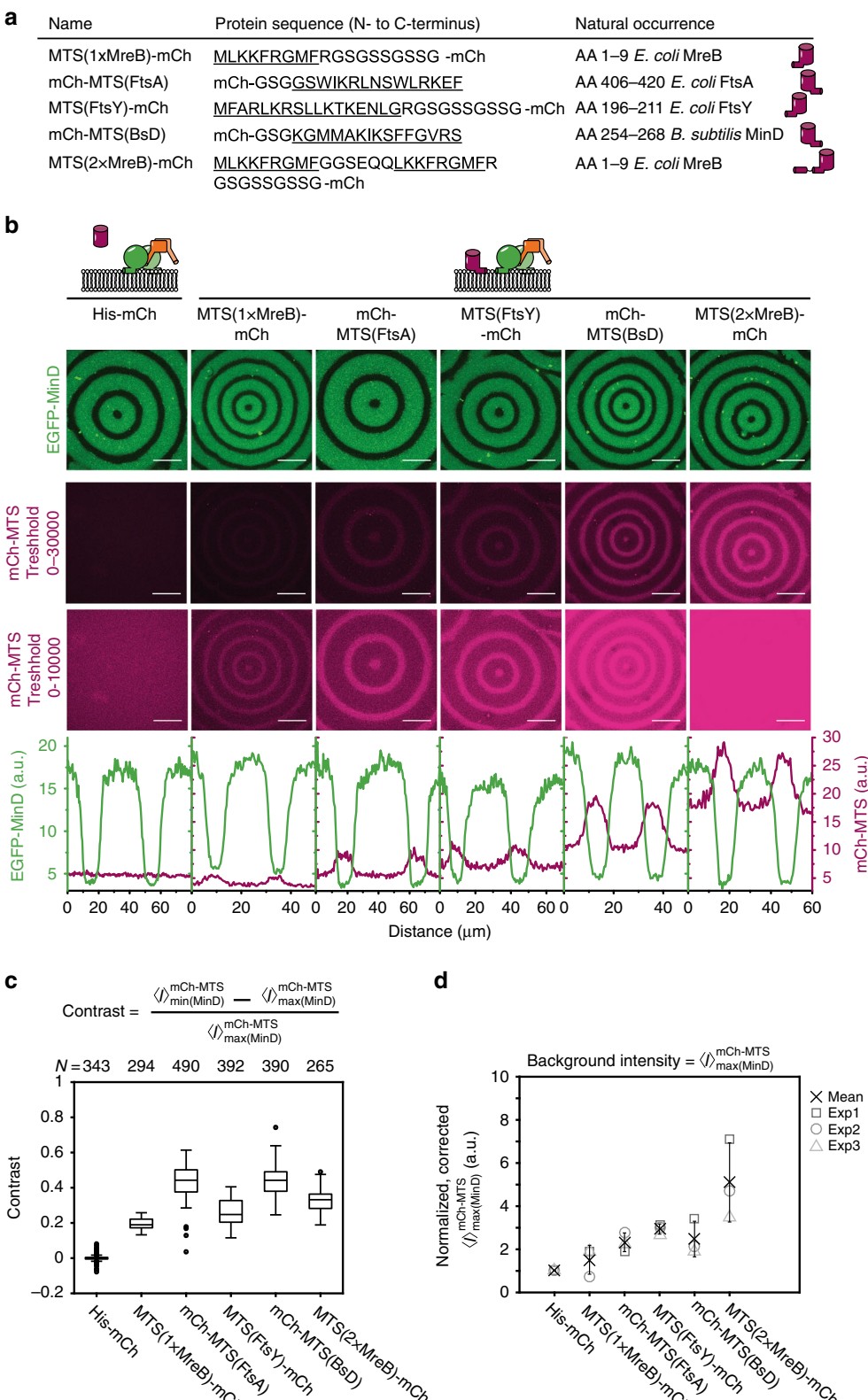

not affected by mCh-MTS constructs: average MinD intensities on the membrane (Supplementary Fig. 3d-f) and wavelength and velocity of MinDE waves were similar in the presence of all mCh-MTS constructs and the control His-mCh (Supplementary Fig. 4). Spatiotemporal positioning of the strongest mCh-MTS, MTS(2×MreB)-mCh, is robust, as it occurred for all tested MinD/MinE ratios (10–0.1) (Supplementary Fig. 5a, b), for all mCh-MTS/MinDE ratios, as high as 30 and as low as 0.1 (Supplementary Fig. 6), and also at the lowest equimolar MinDE concentration that still supported self-organization in our assay (MinDE = 0.4 μM) (Supplementary Fig. 5c-e), albeit with varying strength.

**Fig. 2** MinDE regulate a variety of peripheral membrane proteins to different extents. **a** Overview of the model peripheral membrane proteins employed. All amphipathic helices were fused to mCherry at their endogenous terminus. **b** Representative images of the MinDE wave (upper panel) and the anticorrelated mCh-MTS wave with two different brightness settings (middle and lower panels) on the membrane (1 μM mCh-MTS, 1 μM MinD (30% EGFP-MinD), 1 μM MinE). All images in one row were acquired and displayed using the same instrumental settings. Fluorescence intensity line plots of the corresponding images (EGFP-MinD fluorescence in green, mCh-MTS fluorescence in magenta) show the difference in the extent of the spatial regulation (lowest panel). **c** mCh-MTS constructs with a C-terminal amphipathic helix exhibit highest contrast. Box plot of the contrast of mCh-MTS constructs, lines are median, box limits are quartiles 1 and 3, whiskers are 1.5× interquartile range (IQR) and points are outliers. **d** mCh-MTS intensity in the MinDE maximum ($I_{\mathrm{max(MinD)}}^{\mathrm{mCh-MTS}}$) normalized to His-mCh and corrected for the fluorescent protein fraction. Each data point (square, sphere, triangle) corresponds to one independent experiment (exp 1–3) and was generated from at least one tile scan (7 by 7) in one sample chamber (number of images $N_{\mathrm{His-mCh}} = 343$, $N_{\mathrm{MTS(1\times MreB)-mCh}} = 294$, $N_{\mathrm{mCh-MTS(FtsA)}} = 490$, $N_{\mathrm{MTS(FtsY)-mCh}} = 392$, $N_{\mathrm{mCh-MTS(BsD)}} = 390$, $N_{\mathrm{MTS(2\times MreB)-mCh}} = 265$). Cross and error bars represent the mean value and standard deviation of the three independent experiments. Scale bars: 50 μm

Next, we designed two constructs harboring two copies of the *E. coli* MinD MTS, mCh-MTS(2×MinD) and mCh-Jun-MTS(1×MinD), which both strongly bound to the membrane (Supplementary Fig. 1a, c). Intriguingly, they were also efficiently regulated by MinDE (Supplementary Fig. 7, Supplementary Movie 3). These two proteins should have a similar membrane affinity as the alleged MinD species on the membrane, a MinD dimer. This suggests that MinDE membrane binding involves higher-order recruitment or oligomerization. Membrane binding of mCh-MTS (1×MinD), containing a single copy of the *E. coli* MinD MTS, could not be detected in agreement with previous reports[16] (Supplementary Fig. 1a, c). Thus regulation by MinDE was also negligible (Supplementary Fig. 7).

Taken together, we show that MinDE spatiotemporally regulates model peripheral membrane proteins over a wide range of concentrations through a nonspecific mechanism independent of the specific amphipathic helix employed. This regulation can be rationalized by a competition for membrane binding sites between MinDE and mCh-MTS constructs.

**MinDE is a spatial sorter for lipid-anchored proteins.** Next, we asked whether MinDE dynamics could also regulate proteins that, unlike mCh-MTS constructs, are unable to dissociate from the membrane, similar to transmembrane proteins in vivo. Full transmembrane proteins are static on SLBs, because they are in contact with the support[35]. Thus, to mimic a diffusible transmembrane protein in our assays, we used Alexa647-labeled streptavidin coupled to biotinylated lipids in the SLB (Fig. 3b). The tetrameric streptavidin binds two to three biotinylated lipids simultaneously rendering the dissociation of streptavidin negligible on the timescale of the MinDE waves, while the lipid-anchoring ensures diffusive mobility in the membrane[39]. The resulting streptavidin membrane density was about $6.6 \times 10^3/\mu m^2$ and, assuming a streptavidin footprint of 25 nm², covers about 17% of the total available membrane area (Supplementary Fig. 12)[39]. Upon initiating MinDE self-organization by ATP addition, an anticorrelated, directional movement of streptavidin was observed (Fig. 3a, c, Supplementary Movie 4, Supplementary Fig. 8). The kymograph of streptavidin movement differed from those obtained with mCh-MTS constructs in that streptavidin amassed in MinDE minima (compare Figs. 3c and 1b). Even more strikingly, over time streptavidin accumulated in areas where MinDE waves were colliding, or at the edges of MinDE spirals, whereas centers of MinDE spirals were depleted in streptavidin after longer incubation (Fig. 3d, Supplementary Fig. 8). Fluorescence intensity line plots through these stable spirals revealed that streptavidin depletion is correlated with MinDE enrichment and vice versa (Fig. 3d). Regulation of streptavidin and gradient formation was also evident if non-labeled MinDE were used or when dyes were exchanged to mRuby3-MinD and Alexa488-streptavidin (Supplementary Fig. 2b, c). Hence, MinDE self-organization establishes directional mass transport into large-scale streptavidin gradients on the membrane, beyond streptavidin

merely following the MinDE pattern. To test whether this gradient formation is reversible and maintained by the MinDE self-organization, and not due to other effects such as streptavidin 2D crystal formation[40], we used sodium orthovanadate (Na₃VO₄), a generic, competitive phosphatase inhibitor. Addition of Na₃VO₄ to an assay with established MinDE spirals led to MinDE detachment from the membrane (Fig. 3e), followed by the equalization of small scale streptavidin patterns within seconds. The large-scale streptavidin gradients disappeared only after several hundreds of seconds, reestablishing a homogenous distribution (Fig. 3e, Supplementary Movie 5, Supplementary Fig. 8). Hence, MinDE self-organization spatiotemporally regulates membrane-bound streptavidin, establishing directional mass transport and maintaining large-scale concentration gradients. In summary, MinDE self-organization represents a molecular sorting system for membrane-anchored molecules.

**MinDE regulate proteins when mimicking in vivo conditions.** Having found that MinDE regulates unrelated proteins in vitro, we asked whether this could be a relevant phenomenon in vivo. First, we confirmed that MinDE also spatiotemporally regulate both mCh-MTS(BsD) and lipid-anchored streptavidin on membranes made from *E. coli* polar lipid extract (Supplementary Fig. 9). Second, as MinC is an integral part of the MinCDE system in vivo, we showed that the regulation of mCh-MTS (BsD) and lipid-anchored streptavidin is independent of MinC addition (Supplementary Fig. 10). Third, we determined the membrane densities of MinD and MTS(2×MreB)-mCh for four different MinDE concentrations using Fluorescence Correlation Spectroscopy based image calibration (Supplementary Figs. 11, 12, Supplementary Methods). At standard conditions ([MinDE] = 1 μM), peak MinD-membrane densities were high ($1.3 \times 10^4/\mu m^2$) (Supplementary Fig. 13). However, when protein concentrations were lowered to the limit where self-organization still occurred ([MinDE] = 0.4 μM) (Supplementary Fig. 5c-e), peak MinD densities ($1.8 \times 10^3/\mu m^2$) were similar to MTS(2×MreB)-mCh membrane densities ($2.0–2.6 \times 10^3/\mu m^2$) and on the same order of magnitude as the estimated in vivo densities of about $1 \times 10^3/\mu m^2$ (Supplementary Fig. 13, Supplementary Note 1). Hence, our in vitro assay allows to observe spatiotemporal positioning of membrane proteins by MinDE without the interference of a complex cellular environment, while keeping central conditions comparable to the in vivo situation.

**MinDE induce generic protein gradients in microcompartments.** We subsequently visualized MinDE pole-to-pole oscillations in rod-shaped microcompartments in the presence of the two model membrane proteins, mCh-MTS(BsD) and lipid-anchored streptavidin (Fig. 4a). In the past, we demonstrated that confinement of MinDE in microcompartments leads to

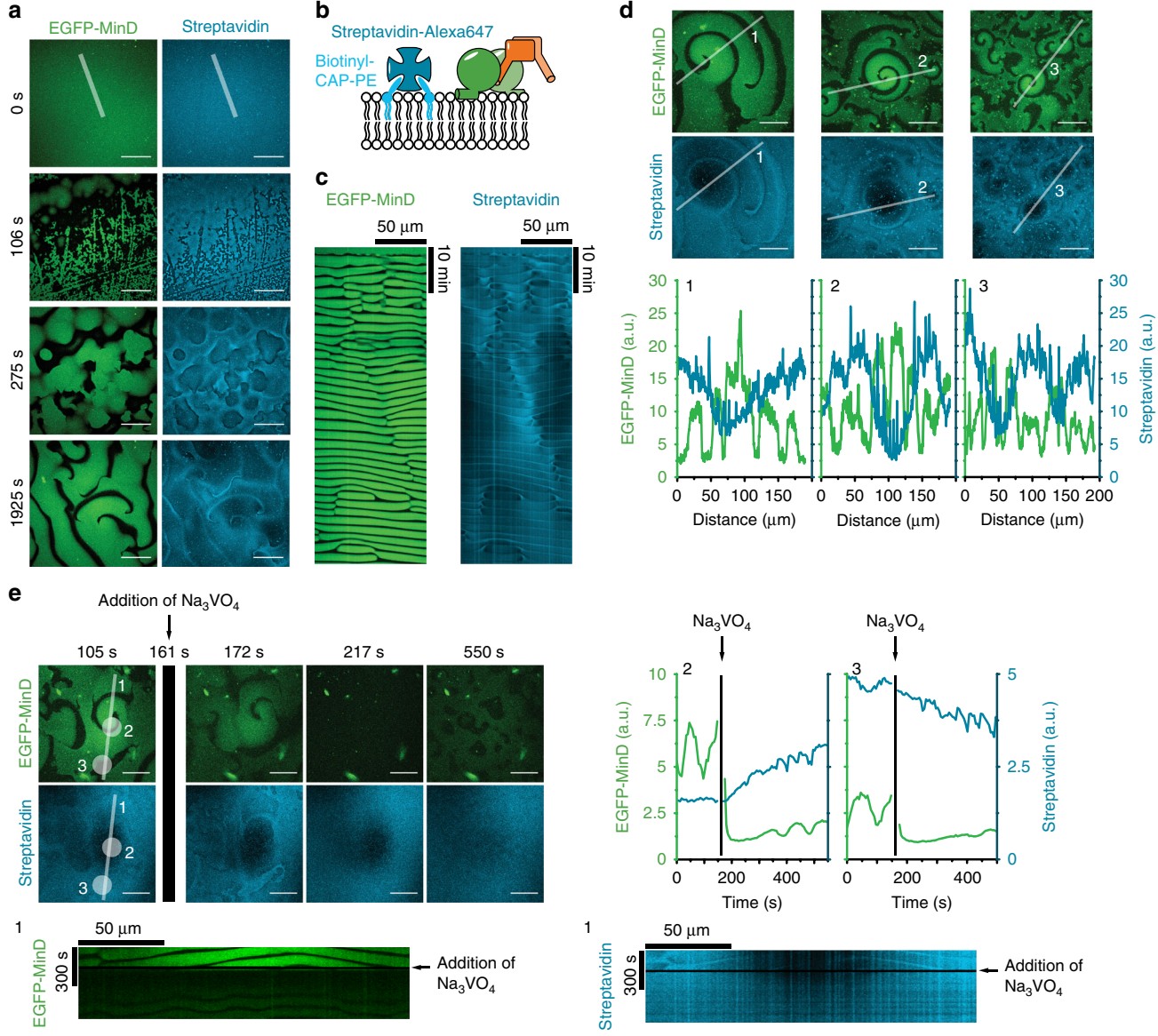

**Fig. 3** MinDE spatiotemporally position a lipid-anchored protein resulting in large-scale concentration gradients. **a** MinDE self-organization spatiotemporally regulates lipid-anchored streptavidin. Representative time series of MinDE self-organization on a SLB with Biotinyl-CAP-PE-bound streptavidin (1 μM MinD, 1 μM MinE, streptavidin-Alexa647). ATP is added at $t = 0$ s to start self-organization. Scale bars: 50 μm. **b** Schematic of the experimental setup. Tetrameric streptavidin is anchored to the SLB by binding two to three Biotinyl-CAP-PE lipids and MinDE and ATP are added. **c** Kymograph of the line selections shown in **a**. Scale bars: 50 μm and 10 min. **d** MinDE self-organization leads to large-scale concentration gradients of streptavidin. Representative images of streptavidin distribution in MinDE spirals after >1 h of MinDE self-organization on SLBs. Fluorescence intensity line plots of EGFP-MinD and streptavidin distribution of selections shown in the respective images. Scale bars: 50 μm. **e** Large-scale streptavidin gradient formation by MinDE is reversible. Representative images and kymograph (1) of a running MinDE assay in the presence of anchored streptavidin. Addition of sodium orthovanadate ($Na_3VO_4$) leads to MinDE detachment which in turn leads to homogenization of streptavidin fluorescence on the membrane. Fluorescence intensity of streptavidin (cyan) and EGFP-MinD (green) is plotted over the duration of the time-lapse in the center (2) and at the rim of the MinDE spiral (3). Scale bars: 50 μm and 300 s. All experiments were performed independently three or more times under identical conditions

pole-to-pole oscillations similar to the observations in vivo[10], albeit on a larger length scale. Indeed, MinDE spatially regulated streptavidin and mCh-MTS(BsD) in microcompartments (Fig. 4b, e, Supplementary Movie 6, 7). Similar to the behavior of the different protein waves on large SLBs, the difference in the resulting counter-oscillations in microcompartments was also evident. Whereas mCh-MTS(BsD) fluorescence was decreased in MinDE occupied areas, but was otherwise homogenously distributed, the streptavidin fluorescence accumulated at the rear of

the MinDE wave (Fig. 4b, e). When MinDE oscillate from pole-to-pole in vivo, a time-averaged concentration gradient of MinD is established. MinC, antagonist of FtsZ assembly that passively follows MinD oscillations, shows the same time-averaged concentration gradient. We analyzed the time-averaged concentration gradient of MinD and, in agreement with our previous study[10], in both cases the MinD concentration showed the characteristic profile with maxima at the compartment poles and minima at mid-compartment, featuring a pronounced dip

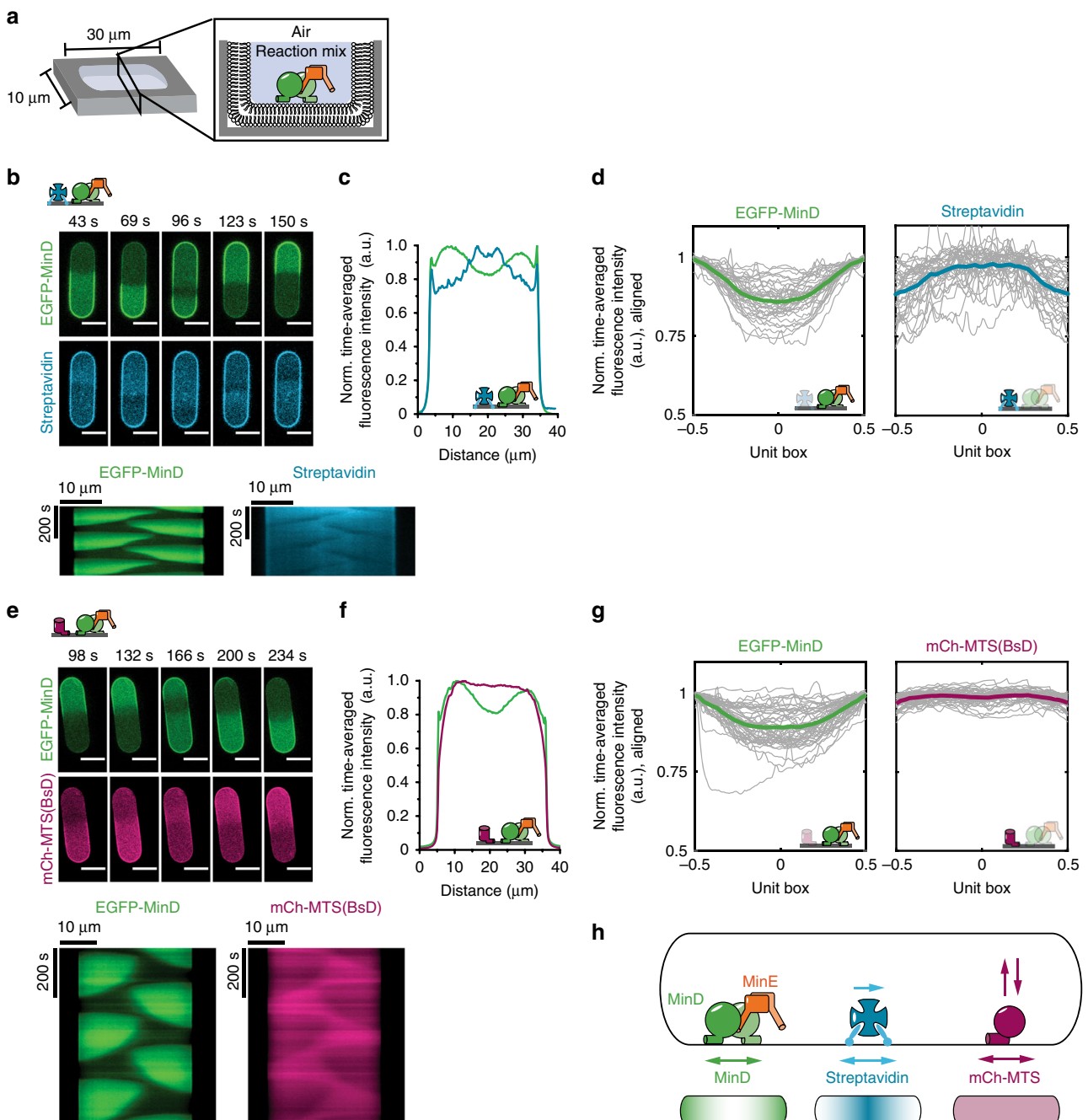

**Fig. 4** MinDE induce oscillatory and time-averaged concentration gradients of model membrane proteins in microcompartments. **a** Experimental setup: PDMS-microcompartments are lined with an SLB and covered by air to confine the proteins. **b** Representative time-lapse images and kymographs of MinDE oscillations and streptavidin counter-oscillations in the compartments (1 μM MinD, 2 μM MinE, streptavidin-Alexa647). Brightness of the streptavidin channel was corrected for bleaching using histogram matching in Fiji. Scale bars: 10 μm. **c** Time-averaged fluorescence intensity profiles of MinDE (green) and streptavidin (cyan) oscillation in **b** showing clear concentration gradients for both MinD and streptavidin. **d** Time-averaged fluorescence intensity profiles (gray lines) for EGFP-MinD and streptavidin aligned and projected to a unit box (see Supplementary Fig. 14 for details). Bold, colored lines represent the mean profiles, generated from three independent experiments with N = 35 microcompartments. **e** Representative time-lapse images and kymographs of MinDE oscillations and mCh-MTS(BsD) counter-oscillations in PDMS microcompartments (1 μM MinD (30% EGFP-MinD), 2 μM MinE, 0.5 μM mCh-MTS(BsD)). Scale bars: 10 μm. **f** Time-averaged fluorescence intensity profiles of MinDE (green) and mCh-MTS(BsD) (magenta) oscillations in **e** showing a clear protein gradient for MinD and homogenous protein distribution of mCh-MTS(BsD). **g** Time-averaged fluorescence intensity profiles (gray lines) for EGFP-MinD and mCh-MTS(BsD) aligned and projected to a unit box. Bold, colored lines represent the mean profiles, generated from three independent experiments with in total N = 45 microcompartments. **h** Schematic explaining how the MinDE system positions lipid-anchored streptavidin and mCh-MTS constructs in rod-shaped microcompartments. MinDE oscillations drive counter-oscillations of lipid-anchored streptavidin and mCh-MTS constructs, thereby establishing a time-averaged concentration gradient of lipid-anchored streptavidin with maximal concentration in the geometric center, but no concentration gradient of mCh-MTS

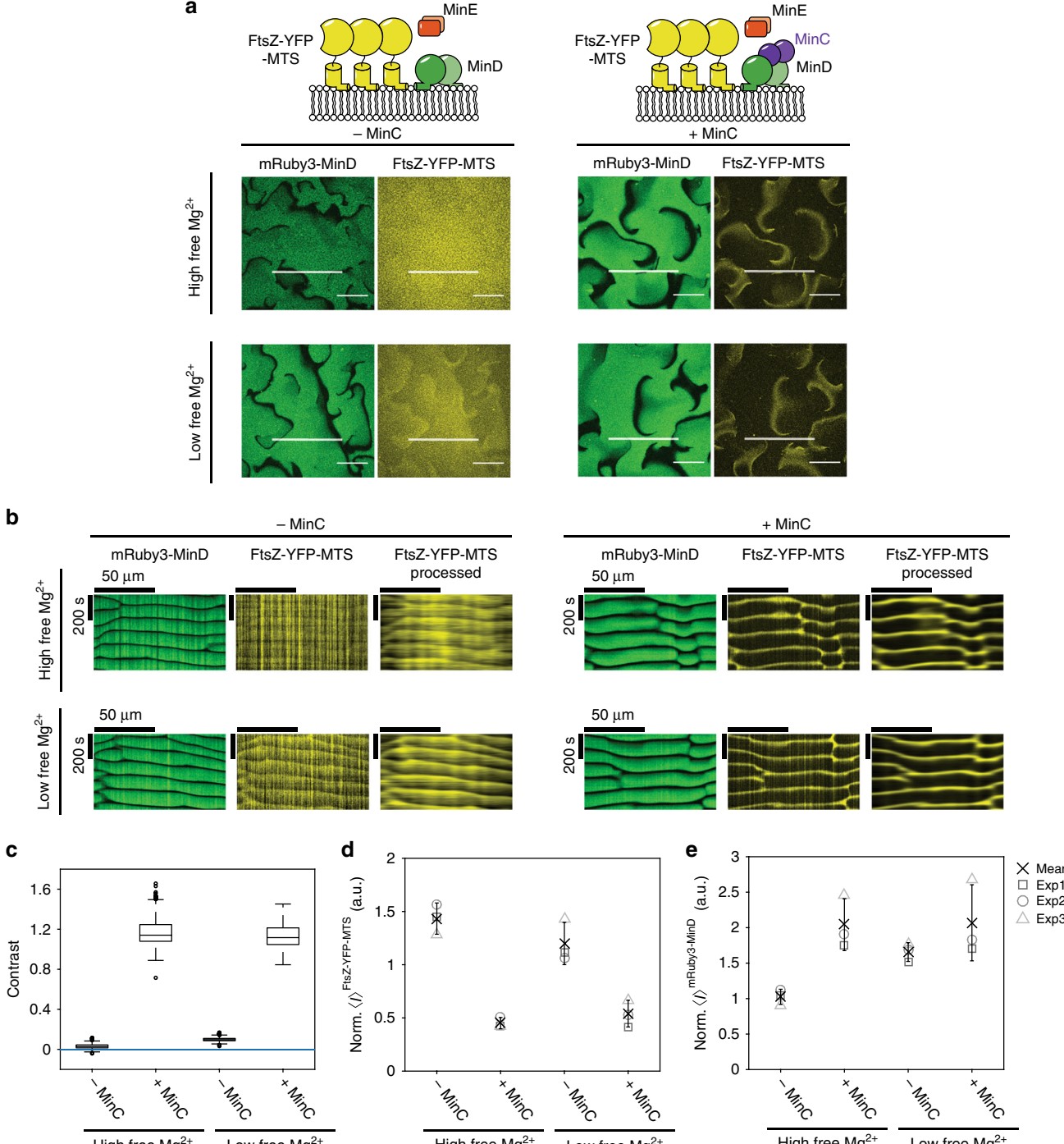

**Fig. 5** MinC enhances spatiotemporal regulation of FtsZ-YFP-MTS by MinDE. **a** Representative images of MinDE self-organization in the presence of FtsZ-YFP-MTS with high and low free $Mg^{2+}$ (~5 and ~1 mM $Mg^{2+}$) and with and without MinC (1 μM MinD (30 % EGFP-MinD), 1 μM MinE, 0.5 μM FtsZ-YFP-MTS, with and without 0.05 μM MinC) corresponding to the timepoint of 6.5 min. All images of the same spectral channel were acquired and displayed using the same instrumental settings. **b** Kymographs of the line selections shown in **a**. The kymograph for FtsZ-YFP-MTS is displayed for unprocessed images (middle panels) and preprocessed images (see Methods) (right panels). **c** MinC increases the regulation of FtsZ-YFP-MTS. Box plot of the contrast of FtsZ-YFP-MTS, lines are median, box limits are quartiles 1 and 3, whiskers are 1.5× IQR and points are outliers. Blue line marks no difference between the intensities in the minima and maxima of the MinDE wave (zero contrast). **d** Average FtsZ-YFP-MTS intensity of the full image normalized to a fluorescent standard. **d** Average mRuby3-MinD density of the full image normalized to a fluorescent standard. Each data point (square, sphere, triangle) (exp 1–3) was generated from one time series consisting of 150 frames. Cross and error bars represent the mean value and standard deviation of the three independent experiments

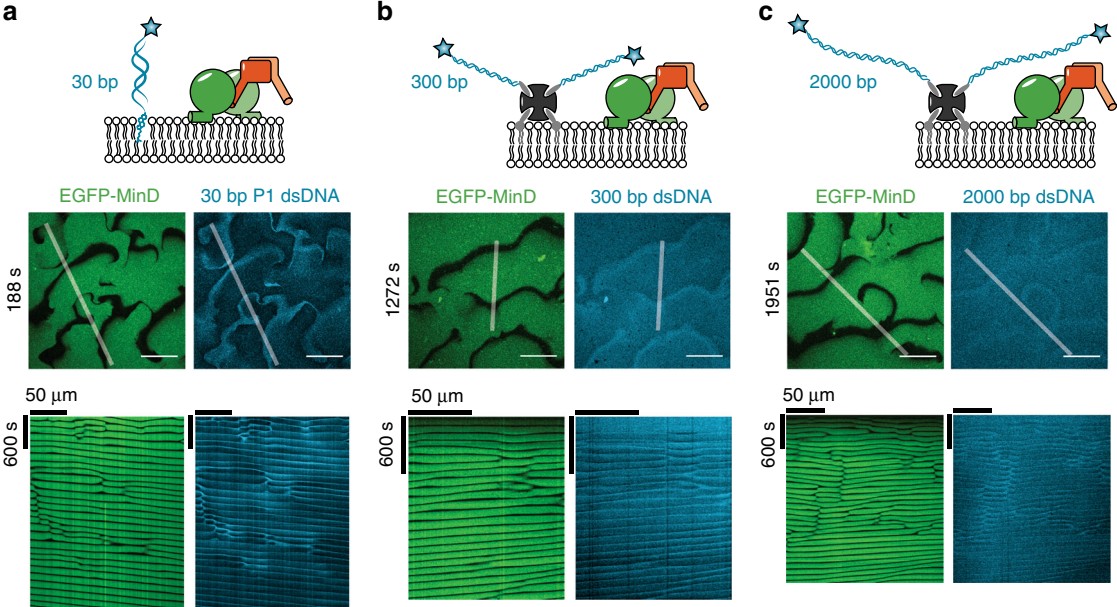

**Fig. 6** MinDE spatiotemporally regulate membrane-anchored DNA. **a** MinDE self-organization can regulate short membrane-anchored DNA fragments. Representative images and kymograph of a time-series of MinDE self-organization in the presence of 30 bp P1 dsDNA bound to the membrane by a cholesterol anchor (1 µM MinD (30% EGFP-MinD), 1 µM MinE, 10 nM TEG-cholesterol-dsP1). **b** Representative images and kymograph of a time-series of MinDE self-organization spatiotemporally regulating 300 bp long dsDNA bound to lipid-anchored streptavidin (1 µM MinD (30% EGFP-MinD), 1 µM MinE, 300 bp lambda DNA, streptavidin). **c** Representative images and kymograph of a time-series of MinDE self-organization spatiotemporally regulating 2000 bp long dsDNA bound to lipid-anchored streptavidin. All experiments were performed independently two (**c**) or three (**a**, **b**) times under similar conditions. Scale bars: 50 µm

(average profile depth of 0.61 ± 0.28 and 0.44 ± 0.24, compare to Supplementary Fig. 14) (Fig. 4c, d, f, g). We further analyzed the time-averaged concentration profiles of the respective model membrane proteins. Streptavidin showed a clear time-averaged concentration profile with a negative profile depth of −0.42 ± 0.35, indicating enrichment in the middle of the microcompartment (Fig. 4c, d, Supplementary Fig. 14). In contrast, the time-averaged concentration of mCh-MTS(BsD) was almost homogenous along the long axis of the compartment (average profile depth of −0.06 ± 0.13) (Fig. 4f, g, Supplementary Fig. 14). Hence, MinDE spatially regulate model peripheral and membrane-anchored proteins in rod-shaped microcompartments and induce steady-state concentration gradients of membrane-anchored proteins with concentration maxima at mid-compartment (Fig. 4h).

We propose this to be of relevance in *E. coli*. So far, division site selection by the MinCDE system was considered to only depend on the inhibitory action of MinC on FtsZ. However, FtsZ does not bind to the membrane by itself, but via two distinct membrane anchors, ZipA and FtsA. ZipA, a single-pass transmembrane protein[41], cannot be reconstituted on SLBs preserving its mobility[35]. Hence, it is also not regulated by MinDE in vitro[35], but could potentially be enriched at midcell by MinDE in vivo. FtsA, in turn, is binding to the membrane via its C-terminal amphipathic helix[37], and would thus be expected to behave like its corresponding mCh-MTS construct mCh-MTS(FtsA) (Fig. 2b). However, while full-length FtsA was regulated by MinDE, the kymograph of the regulation was more similar to lipid-anchored streptavidin than to mCh-MTS (FtsA) (compare Supplementary Fig. 15, Supplementary Movie 8 to Figs. 1b and 3c). This suggests that FtsA rather behaves like a permanently membrane-attached protein than a monomeric peripheral membrane protein, which is in agreement with previous reports of FtsA oligomerization[42–44]. Hence, we propose that FtsA is counter-oscillating to MinDE in vivo, and depending on the oligomerization state, would possibly be enriched at midcell.

**MinC enhances MinDE-dependent regulation of FtsZ-YFP-MTS.** The capability of MinDE to enrich permanently anchored proteins in the middle of rod-shaped compartments and to position the FtsZ anchor FtsA, opened the question for the additional role of MinC. In vivo MinC is strictly required for correct placement of the division site to prevent a minicell phenotype[7,15] and inhibits FtsZ polymerization in vivo and in vitro[19–21]. Nevertheless, slow FtsZ oscillations were shown to depend on MinCDE oscillations[45] and FtsZ treadmilling dynamics in vivo were slightly altered in Δ*minCDE* strain, but not in a Δ*minC* strain[46]. To evaluate whether MinDE alone can influence FtsZ dynamics and what contribution MinC has in the positioning of FtsZ, we revisited previous experiments, i.e., the co-reconstitution of MinCDE with FtsZ-YFP-MTS. FtsZ-YFP-MTS is a chimeric protein, consisting of the fluorescent protein YFP, a truncated *E. coli* FtsZ (1–366) and the MinD MTS[47]. This protein binds to the membrane without its adaptor proteins, FtsA and ZipA, greatly simplifying the system. Under high free $Mg^{2+}$ conditions, FtsZ-YFP-MTS forms thick, treadmilling filament bundles[10,20,48], whereas under low free $Mg^{2+}$ conditions[48], it forms dynamic chiral vortices similar to native FtsZ and FtsA[49]. Hence, we co-reconstituted MinDE and FtsZ-YFP-MTS with high and low free $Mg^{2+}$ concentration and in the presence or absence of MinC. The spatiotemporal regulation of thick FtsZ-YFP-MTS bundles formed at high free $Mg^{2+}$ in solution (~5 mM $Mg^{2+}$) was hardly detectable if only MinDE were present in the assay, but very strong when MinC was supplied, in agreement with our previous reports[10,20]. (Fig. 5a, b, Supplementary Movie 9). In the case of low free $Mg^{2+}$ (~1 mM $Mg^{2+}$), FtsZ-YFP-MTS forms dynamic rings that were also visibly regulated by MinDE alone, but also here the regulation was drastically increased in the presence of MinC (Fig. 5a, b, Supplementary Movie 9). The contrast of the FtsZ-YFP-MTS regulation (Fig. 5c) increased with the amount of FtsZ-YFP-MTS on the membrane (Fig. 5d) and decreased with the amount of MinD on the membrane (Fig. 5e).

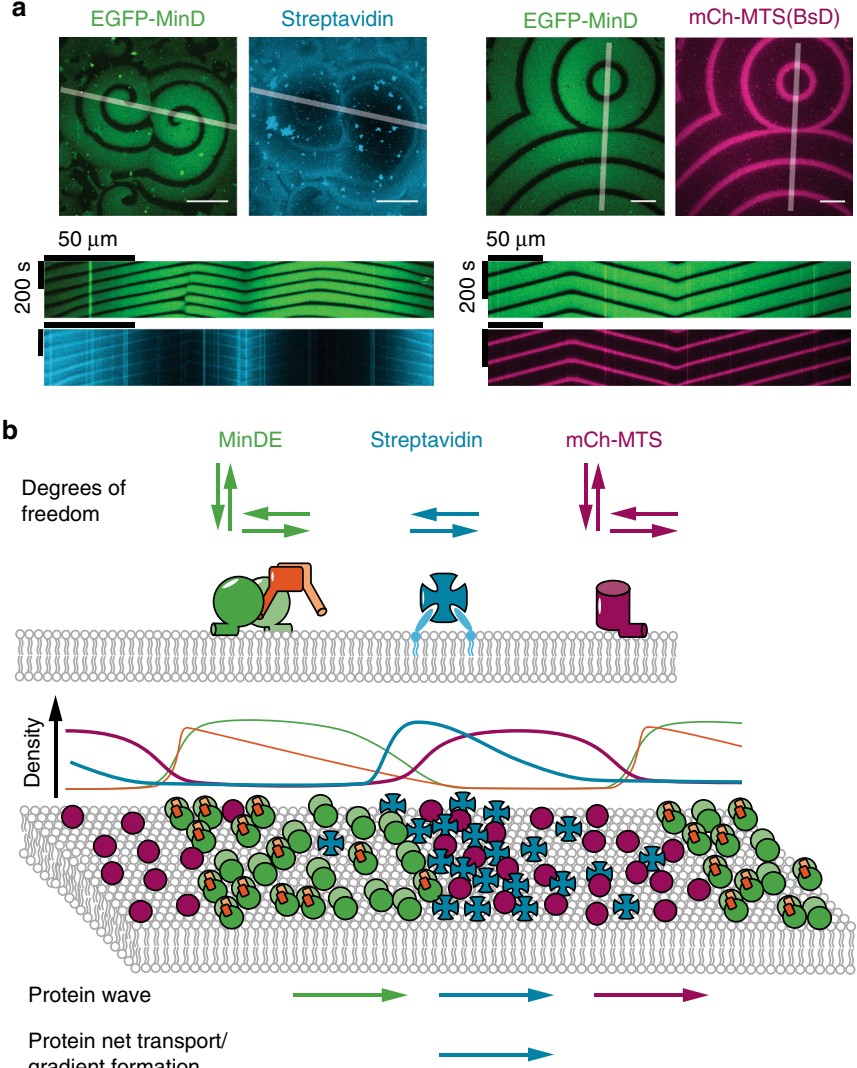

**Fig. 7** MinDE-driven dynamics of model membrane proteins in vitro suggest that MinDE form a propagating diffusion barrier. **a** Representative images and kymographs of colliding MinDE waves in the presence of mCh-MTS(BsD) and lipid-anchored streptavidin bound to biotinylated lipids (1 µM MinD (30% EGFP-MinD), 1 µM MinE, 1 µM mCh-MTS(BsD) or streptavidin-Alexa647). Scale bars: 50 µm. **b** Schematic of the underlying protein behavior resulting in spatiotemporal regulation of model peripheral and membrane-anchored proteins. While mCh-MTS and MinDE can also attach and detach to and from the membrane, streptavidin can only diffuse laterally on the membrane. Schematic density profiles and protein localization on the membrane (magenta: mCh-MTS, green: MinD, orange: MinE, cyan: lipid-anchored streptavidin). The MinDE wave propagates directionally, even if individual proteins show a random movement on the membrane. Both model peripheral and membrane-anchored proteins show a wave propagation in the direction of the MinDE wave. mCh-MTS while more abundant in the MinDE minima covers the membrane homogenously. In contrast the resulting secondary wave of streptavidin shows an inhomogeneous profile and results in a net transport of the membrane-anchored protein

This is consistent with previous results that at high $Mg^{2+}$ concentration, FtsZ-YFP-MTS density on the membrane is higher and individual monomers have a longer residence time on the membrane compared to low free $Mg^{2+}$ concentration[48], while MinC leads to depolymerization of FtsZ-YFP-MTS reducing its density on the membrane[20]. Images acquired with higher magnification showed that neither the large FtsZ-YFP-MTS bundles nor the small dynamic vortices were laterally moved by MinDE, unlike lipid-anchored streptavidin (Supplementary Fig. 16, Supplementary Movie 10). Both, FtsZ-YFP-MTS filaments and dynamic rings, are also not diffusing laterally on the membrane, independent of the presence of MinDE[20,48]. Hence, it can be concluded that for lateral transport of membrane-anchored proteins by MinDE, diffusive mobility is a prerequisite. Instead, FtsZ-YFP-MTS filaments and dynamic vortices varied in intensity in the MinDE minima and maxima suggesting that MinDE are regulating the membrane binding of FtsZ-YFP-MTS. Interestingly, MinDE waves were patterned, showing a negative image of the FtsZ-YFP-MTS filaments and rings (Supplementary Fig. 16), indicating that MinDE can spatiotemporally regulate proteins, even while coupling over small membrane gaps[50] or in this case immobile obstacles.

In summary, we show that MinDE-dependent regulation of FtsZ-YFP-MTS increases with decreasing FtsZ-YFP-MTS density on the membrane and is drastically enhanced by MinC.

**MinD regulates DNA-membrane tethers.** Several studies report a defect in chromosome segregation when MinCDE are deleted in *E. coli*[26–28]. Based on simulations that showed that chromosome segregation can be achieved by static or oscillatory gradients of chromosome-membrane tethers, MinD was suggested to drive chromosome segregation by direct, but transient DNA binding[24].

In light of our results, we hypothesized that any DNA-membrane tether could be spatially regulated by the MinDE system. Different chromosome-membrane contacts have been reported in *E. coli*, such as transertion[30,51], which itself has been suggested to aid chromosome segregation[30].

To model chromosome membrane tethering sites in the most simplistic fashion in our in vitro setup, we employed dsDNA with a fluorescence label and a TEG-cholesterol moiety for membrane binding on opposite ends. As the DNA sequence, we chose a 30-bp-long fragment from the P1 promoter of the *minB* operon in *E. coli*, shown to bind to MinD[24]. When cholesteryl-tagged P1 fragments were included in a MinDE self-organization assay, the oligonucleotides formed traveling waves on the membrane that were anticorrelated to the MinDE waves and not correlated as would be expected for direct DNA binding by MinD (Fig. 6a, Supplementary Fig. 8, Supplementary Movie 11). The P1 fragments showed a phenomenologically similar behavior as the streptavidin-lipid conjugate, accumulating at sites were waves collide. This can be explained by the strong binding of TEG-cholesteryl oligonucleotides to membranes ($K_D = 16$ nM, $k_{off} = 6 \times 10^{-4}$/s)[52], rendering MinDE-induced dissociation unlikely. To rule out that the membrane-anchoring of the DNA mask binding to MinD, we performed control experiments with soluble P1 DNA fragments. We turned to TIRF microscopy (TIRFM) to be able to monitor even transient recruitment of DNA to the MinDE wave. Although the spatiotemporal positioning of membrane-anchored P1 fragment was clearly visible, we could not observe modulation of fluorescence intensity in the DNA channel with soluble P1 fragments under the same settings, nor with increased laser irradiation or DNA concentrations (Supplementary Fig. 17). Thus, we could not detect any recruitment or binding of DNA to MinDE waves. While these experiments cannot fully rule out that DNA binds to MinD, this interaction would be very weak.

If MinDE would indeed participate in chromosome segregation by regulating DNA-membrane tethers, the system needed to be capable of transporting larger cargo. Thus, we performed experiments with longer DNA molecules bound to streptavidin as a spacer. The 300 and 2000 bp long DNA strands were labeled with Cy5-fluorophore and biotin for immobilization on lipid-anchored streptavidin (mass: 185 kDa and 1.2 MDa, contour length: 100 and 640 nm). MinDE spatially organized the streptavidin-bound DNA in both cases (Fig. 6b, c, Supplementary Fig. 8, Supplementary Movie 12, 13). In summary, our results show that MinD is unlikely to bind DNA directly, but MinDE are able to regulate DNA-membrane tethers in vitro.

## Discussion

Here we showed that the MinDE reaction–diffusion system can dictate the localization of membrane proteins in a spatiotemporal manner without specific molecular interactions in vitro. These proteins apparently establish a generic, nucleotide-dependent transport mode for membrane-bound diffusive molecules based on a moving diffusion barrier. This in turn implies a more fundamental role of MinDE in division site selection and chromosome segregation in vivo and may in the future be applied to position and transport molecules in artificial cells.

The spatiotemporal positioning by MinDE in vitro is independent of (1) the target's membrane anchoring (amphipathic helix, lipid anchor or cholesterol anchor), (2) the nature of the target (protein or DNA), (3) the oligomerization state of the target protein, and (4) the target's dwell-time on the membrane (transiently or permanently bound). We hence conclude that MinDE can act as a generic spatial cue for the distribution of functionally unrelated membrane-bound molecules in vitro.

While sensitive to aspects such as diffusive mobility and membrane dwell-time of the regulated components, the observed MinDE-mediated dynamics likely reflect a common underlying mechanism. Comparing the spatiotemporal regulation of the two model membrane proteins that represent the distinct cases, transiently bound mCh-MTS and permanently attached lipid-anchored streptavidin, a stark difference in the effect is evident (Fig. 7a, Supplementary Movie 14). MinDE cannot induce large-scale gradients of mCh-MTS, suggesting that MinDE locally affect the attachment and/or detachment of these proteins by competition for membrane binding sites (Fig. 7b), explaining the overall reduction of mCh-MTS density on the membrane in the presence of MinDE. In contrast, MinDE induce large-scale gradients of the lipid-anchored streptavidin that can only laterally diffuse on the membrane. Hence, the moving MinDE wave front must lead to a directionally biased diffusion of streptavidin on the membrane and thus induce a net protein transport (Fig. 7b). Oligomerized peripheral membrane proteins, such as FtsA, have increased membrane dwell-times and are thus similarly regulated as permanently anchored proteins. In all cases, lateral mobility is a prerequisite for being positioned by MinDE. Consequently, static FtsZ-YFP-MTS networks, which do not diffuse on the membrane, are not subject to a net transport by MinDE. However, a weak, MinDE-induced regulation of protein abundance can be observed.

Further, MinDE drove counter-oscillations of both mCh-MTS(BsD) and lipid-anchored streptavidin in rod-shaped microcompartments. However, only the regulation of streptavidin resulted in a steady-state concentration gradient, where the protein was enriched at midcell. Hence, gradient formation in microcompartments is related to the occurrence of large-scale gradients on planar SLBs.

Based on these observations, we propose that MinDE surface waves constitute a propagating diffusion barrier. Although individual MinDE proteins do not move in a directed fashion on the membrane, but simply attach and detach[9,18], the MinDE wave front as a whole translocates directionally. This sliding concentration wave forms a mobile diffusion obstacle that directionally biases the diffusion of tightly membrane-attached proteins and outcompetes other peripheral membrane proteins during attachment. The detailed biophysical features of this nonspecific molecular transport process will be subject of further investigations.

MinDE membrane binding is a highly cooperative process[53–55], which is commonly attributed to the monomer-dimer transition of MinD during ATP binding[16]. However, MinD, sometimes in conjunction with MinC, was shown to assemble into higher-order structures on the membrane, similar to a 2D filament network[55–58]. Strikingly, MinDE also regulated mCh-MTS(2×MinD) and the dimerizing mCh-Jun-MTS(1×MinD) with similar membrane affinity as a MinD dimer (Supplementary Fig. 7), further corroborating the existence of higher-order recruitment or oligomerization during MinD-membrane binding.

Additionally, MinDE have been shown to modify membranes and to preferentially bind to anionic lipids (Supplementary Note 2)[53,57]. It will thus be interesting to investigate to what extent MinDE surface waves alter the local membrane properties, like viscosity or lipid content, to control attachment and diffusion of other membrane proteins in a more direct way.

Irrespective of the mechanistic details, the mobile diffusion barrier generated by MinDE in vitro is reminiscent of the rather static actin cortex in eukaryotes and the circumferentially rotating actin homolog MreB in bacteria, known to organize lipid domains and regulate protein diffusion[59–61]. We therefore propose that this mechanism is also relevant in vivo. The sole purpose of the MinDE oscillations was long assumed to be the positioning of the

FtsZ-inhibitor MinC, although it seemed counterintuitive that such an energy-consuming process would not be utilized more efficiently by the cell. Several studies provided hints that MinDE oscillations influence chromosome segregation and the distribution and abundance of membrane proteins in vivo (Supplementary Table 1)[25–28,46,62,63]. However, a differentiated and unbiased analysis in vivo remains challenging because: (1) MinCDE manipulations cause cell division defects; (2) observation of membrane dynamics in bacteria is complicated due to their small size, unfavorable optical properties and insufficient labeling strategies. Circumventing these problems, our in vitro assay plays to the strength of a reduced-complexity approach, allowing us to probe the influence of MinDE dynamics on membrane-bound proteins without the interference of a complex cellular environment. This reduction in complexity entails that MinDE self-organization in vitro and in vivo differ with respect to MinDE membrane densities and wavelength. Nevertheless, we confirmed that MinDE-mediated positioning of membrane proteins also occurs when the in vitro assay is closely mimicking in vivo conditions (Supplementary Figs. 9, 10, 13).

Consequently, MinDE might also drive counter-oscillations of membrane proteins in E. coli. Considering the results obtained here, it is, however, unlikely that all inner membrane proteins would be regulated. Proteins that are not freely diffusing on the membrane, such as streptavidin crystals (Fig. 7a, Supplementary Movie 14) or thick FtsZ-YFP-MTS bundles (Supplementary Fig. 16), cannot be moved laterally by the MinDE system. Hence, any protein that is anchored to the cell wall or whose diffusion is confined, e.g., by interaction with MreB filaments[59], would be exempt. Furthermore, depending on the exact mechanism of regulation, transmembrane proteins with none or small cytosolic domains, or proteins that favor a certain lipid composition, e.g. anionic phospholipids, may not be regulated or subject to other, stronger spatial cues.

Proteins potentially organized by MinDE in vivo include monomeric peripheral membrane proteins, whose abundance would be decreased by MinDE (Supplementary Fig. 18). This MinDE-induced decrease has already been observed for several such proteins in vivo, among them FtsY whose corresponding mCh-MTS construct was also regulated in vitro (Fig. 2, Supplementary Table 1)[25].

Furthermore, we did not detect direct DNA binding of MinD in our assay, which was proposed to explain the chromosome segregation defects occurring in ΔminCDE strains (Supplementary Table 1)[24,26–28]. Instead we suggest that MinDE, as the closest homolog of ParABS systems mediating chromosome segregation in other bacteria, influence the spatiotemporal organization of DNA-membrane tethers, which are manifold in E. coli[30,51].

Finally, regulation of mobile transmembrane proteins and strongly membrane-bound oligomeric proteins, would result in their enrichment at midcell. The two FtsZ anchors, ZipA and FtsA, represent such protein classes and hence would be pre-positioned at the future division site (Supplementary Fig. 18). Indeed, ZipA and MinCDE counter-oscillate in vivo, so far accredited to recruitment of ZipA to FtsZ, that is periodically depolymerized by MinC[64]. Several other observations also point towards such a mechanism (Supplementary Table 1). The presence of MinC strongly enhances division site selection, as demonstrated by the strong increase in regulation of the chimeric FtsZ-YFP-MTS when MinC was supplied (Fig. 5). However, both processes may be intertwined: MinC depolymerizes FtsZ and hence might free its membrane anchors to be positioned by MinDE.

In summary, the MinDE-dependent regulation of membrane-bound molecules by a propagating diffusion barrier can be seen as an archetypal physicochemical mechanism based on two proteins only. Our data suggest that this reaction–diffusion system is capable of spatially regulating a much larger set of proteins than previously known. Other factors such as MinC augment the system by providing protein specificity.

Without doubt, our in vitro results will motivate future in vivo studies to discern to what extent MinDE is regulating nonspecific spatiotemporal organization of membrane proteins. We speculate that also other reaction–diffusion systems, such as ParABS[1], Cdc42[31], and PAR[32] proteins may be capable of regulating various proteins on their respective matrix. Moreover, our work lays the foundation to apply this simplistic regulatory mechanism for positioning artificial division machineries and chromosomes in constructing a synthetic cell from the bottom up. The ability to control MinDE waves by geometric cues and light will potentially allow a precise and controlled spatiotemporal targeting of any membrane-bound molecule[50,65].

## Methods

**Plasmids**. A list of all plasmids and primers used in this study and their construction can be found in the Supplementary Information (Supplementary Table 2 and 3, Supplementary Methods).

**Protein purification**. Purification of His-MinD, His-EGFP-MinD, His-mRuby3-MinD, His-MinC, and His-MinE was performed essentially as described earlier[9]. For a detailed protocol see Ramm and Glock et al[66]. In brief, proteins were purified via Ni-NTA affinity purification. Protein was further purified using gel filtration chromatography in storage buffer (50 mM HEPES, pH 7.25, 150 mM KCl, 10% Glycerol, 0.1 mM EDTA). Proteins were quick-frozen and stored in aliquots at −80 °C until further use.

Purification of FtsA was performed similar as described earlier[49]. FtsA was expressed as His-SUMO-Gly5 fusion from plasmid pML60[49] in E. coli OverExpress™ C41(DE3) pLysS (Sigma-Aldrich, St. Louis, USA) in autoinduction medium (ZYM5052)[67]. Cells were lysed in lysis buffer (50 mM Tris-HCl pH 8.0, 500 mM NaCl, 10 mM MgCl₂, 10 mM imidazole, 0.4 mM TCEP, 1 mM ADP and 10 mM CHAPS, EDTA-free complete plus protease inhibitor (Roche, Basel, Switzerland), 10 U/ml DNase 1, 100 μg/ml lysozyme) by sonication with a tip sonicator (2.30 min, 30 s pulses, 30% amplitude). After centrifugation to clear cell debris (30 min, 25,000 × g, 4 °C) the lysate was incubated with Ni-NTA agarose (Qiagen, Hilden, Germany) for 30 min. The agarose beads were washed thrice with wash buffer (50 mM Tris-HCl pH 8.0, 500 mM KCl, 10 mM MgCl₂, 20 mM imidazole, 0.4 mM TCEP, 1 mM ADP), subsequently protein was eluted (50 mM Tris-HCl pH 8, 500 mM KCl, 10 mM MgCl₂, 200 mM imidazol, 1 mM CHAPS, 0.4 mM TCEP, 1 mM ADP). Buffer was exchanged to labeling buffer (50 mM HEPES/NaOH pH 7.5, 500 mM KCl, 10 mM MgCl₂, 10 mM CaCl₂, 10% glycerol, 0.4 mM TCEP, 1 mM ADP) using a Econo-Pac 10DG desalting column (Biorad, Hercules, USA). About 50 μM SUMO-Gly5-FtsA was incubated with 1 μM SenP2, 50 μM Sortase A (highly efficient mutant)[68], 0.3 mM Cy5-LPETGG in labeling buffer for 2–3 h. Cy5-LPETGG was produced by solid-phase peptide synthesis using Fmoc chemistry. Cy5-FtsA was separated from SenP2, Sortase, non-reacted peptide and non-cleaved protein by gel-filtration chromatography on a 16/600 Superdex 200 pg column (GE Healthcare, Pittsburgh, USA) equilibrated in storage buffer (50 mM HEPES/NaOH pH 7.5, 500 mM KCl, 10 mM MgCl₂, 10% glycerol, 0.4 mM TCEP, 1 mM ADP). Protein aliquots were quick-frozen and stored at −80 °C.

All mCh-MTS constructs were expressed in E. coli BL21 (DE3) (Agilent Technologies, Santa Clara, USA) in TB medium. Medium was inoculated from an overnight culture and cells were grown to an OD₆₀₀ of 0.5–0.8 at 37 °C. Subsequently cells were induced with 0.5 mM IPTG and shifted to 16 °C for protein expression. After 12–16 h, cells were harvested by centrifugation and cell pellets were stored at −20 °C until further use. For purification, cell pellets were resuspended in lysis buffer (50 mM Tris-HCl pH 8.0, 300 mM NaCl, 10 mM Imidazole, 0.4 mM TCEP, EDTA-free complete plus protease inhibitor (Roche), 10 U/ml DNase 1, 100 μg/ml lysozyme) and lysed by sonication with a tip sonicator (2.30 min, 30 s pulses, 30% amplitude). After centrifugation to clear cell debris (45 min, 25,000 × g, 4 °C), the lysate was incubated with Ni-NTA agarose (Qiagen, Hilden, Germany) for 1 h. Beads were washed with wash buffer thrice (50 mM Tris-HCl pH 8.0, 300 mM NaCl, 20 mM imidazole, 10% glycerol, 0.4 mM TCEP) and subsequently the protein was eluted with elution buffer (50 mM Tris-HCl pH 8.0, 300 mM NaCl, 250 mM imidazole, 10% glycerol, 0.4 mM TCEP). Purity was assessed with SDS-PAGE and buffer was exchanged to storage buffer (50 mM HEPES/NaOH pH 7.25, 150 mM KCl, 0.1 mM EDTA, 10% glycerol, 0.4 mM TCEP) using a Econo-Pac 10DG desalting column (Biorad, Hercules, USA). Aliquots were snap-frozen and stored until further use at −80 °C.

FtsZ-YFP-mts protein was purified as previously described[48] according to a protocol from Osawa et al[47]. Briefly, the protein was expressed in E. coli BL21 (DE3) (Laboratory of German Rivas, CIB, CSIC, Madrid, Spain). Cells were grown

until $OD_{600}$ of 0.8 and then protein expression was induced by addition of 0.5 mM IPTG and cells were shifted to 20 °C. After growth for 14–16 h cells were harvested by centrifugation at 3200 rpm and 4 °C. Subsequently, FtsZ-YFP-MTS was precipitated from the supernatant through 30% ammonium sulfate and a 20 min incubation on ice while slowly shaking. After centrifugation (3200 rpm, 4 °C) and re-suspension of the pellet, the protein was purified by anion exchange chromatography using a 5 × 5 ml Hi-Trap Q-Sepharose column (GE Healthcare, Chicago, USA).

Purity and integrity of all proteins was assessed using SDS-PAGE and mass spectrometry. All protein concentrations were measured using Bradford assay and the fluorescent fraction of every protein was determined by absorption spectroscopy using a V-650 spectrophotometer (Jasco, Pfungstadt, Germany).

**Preparation of supported lipid bilayers.** Coverslides were rubbed and rinsed with EtOH and ddH$_2$O and a plastic ring was glued on top to generate a sample chamber. The slide was further cleaned in a plasma cleaner (model Zepto, Diener electronic, Ebhausen, Germany) for 1 min at 30% power and 0.3 mbar with oxygen as process gas. All lipids were purchased from Avanti Polar Lipids (Alabaster, AL, USA). Small unilamellar vesicles were prepared at a concentration of 4 mg/ml in buffer A (25 mM Tris-HCl pH 7.5, 150 mM KCl, 5 mM MgCl$_2$) for DOPC/DOPG mixtures or buffer B (25 mM Tris-HCl pH 7.5, 150 mM KCl) for E. coli polar lipid extract. Unless otherwise noted the lipid composition was 70 mol % DOPC and 30 mol % DOPG, previously shown to yield similar MinDE behavior as on E. coli polar lipid extract[10,11]. Lipids dissolved in chloroform were dried under a nitrogen stream and vials were placed in a desiccator to remove residual chloroform for at least 30 min. Afterwards lipids were slowly rehydrated in Buffer A or B and SUVs were generated by sonication in a sonicator bath until the solution appeared clear. To generate supported lipid bilayers (SLB) SUVs were added to the reaction chamber at a concentration of 0.5 mg/ml in buffer A for DOPC/DOPG mixtures or buffer C (25 mM Tris-HCl pH 7.5, 150 mM KCl, 3 mM CaCl$_2$) for E. coli polar lipid extract. After 4 min incubation on a 37 °C warm heating block, the SLB was washed 10 times with a total of 2 ml buffer B to remove excess vesicles. Before self-organization assays, the buffer in the chamber was exchanged with reaction buffer (25 mM Tris-HCl pH 7.5, 150 mM KCl, 5 mM MgCl$_2$).

**Self-organization assays.** Self-organization assays were performed similar as described earlier[9,66]. Self-organization assays were performed on preformed SLBs in 200 µl reaction buffer (25 mM Tris-HCl pH 7.5, 150 mM KCl, 5 mM MgCl$_2$) supplemented with 2.5 mM Mg-ATP (stock: 100 mM ATP in 100 mM MgCl$_2$, adjusted to pH 7.5) and at a constant room temperature of 23 °C. MinD and MinE were used at 1 µM protein concentration each unless otherwise noted. MinC if included was used at a final concentration of 0.05 µM. For labeling MinD was doped with 30 mol % of EGFP-MinD/mRuby3-MinD in each case. For experiments with mCh-MTS constructs, all proteins were added to the sample chamber first (order MinD, MinE, (MinC), mCH-MTS) and then the reaction was started by addition of Mg-ATP. Sample chambers were mixed by pipetting, lidded and incubated for 1 h before image acquisition. For z-stack acquisition to determine membrane binding of mCh-MTS constructs, mCh-MTS was incubated at 1 µM final concentration in the absence of MinDE on labeled, preformed SLBs (70 mol % DOPC, 30 mol % DOPG, 0.05 mol % ATTO655-PE) in reaction buffer for more than 1 h before image acquisition.

Experiments with FtsA were conducted at 0.4 µM final concentration in 200 µl reaction buffer 2 (pH 7.5, 12.5 mM Tris-HCl, 25 mM HEPES/KOH, 325 mM KCl, 5% glycerol, 7.5 mM MgCl$_2$). FtsA was added directly after MinDE and before ATP addition, samples were mixed, lidded and reaction was started with 2.5 mM ATP directly before image acquisition.

Self-organization assays in the presence of FtsZ-YFP-MTS were performed in 200 µl reaction buffer. Proteins were added to the chamber first (1 µM MinD (30% mRuby3-MinD), 1 µM MinE, with and without 0.05 µM MinC, 0.5 µM FtsZ-YFP-MTS). Samples were mixed and lidded and self-organization of MinDE and FtsZ-YFP-MTS was started by addition of 2.5 mM Mg-ATP and 0.4 or 4 mM GTP, for high and low free Mg$^{2+}$ concentrations, respectively. Image acquisition was started after 10 min of incubation with ATP and GTP.

**Streptavidin-bound membranes.** All three streptavidin forms used (non-labeled, Alexa647-labeled or Alexa488-labeled) were purchased from ThermoFisher Scientific (Waltham, USA). For experiments involving streptavidin anchored to biotinylated lipids SLBs were prepared as described above with the lipid composition of 69 mol % DOPC/30 mol % DOPG/1 mol % Biotinyl-CAP-PE or E. coli polar lipid extract doped with 1 mol % of Biotinyl-CAP-PE. After formation of SLBs the buffer was exchanged to 200 µl reaction buffer and streptavidin was added at concentration of 1 µg/ml. Chambers were incubated for 30–60 min at room temperature. Subsequently unbound streptavidin was removed from the chambers by washing five times with a total volume of 1 ml reaction buffer. MinD and MinE were added at 1 µM each in a total of 200 µl reaction buffer and the reaction was started directly before imaging by addition of ATP to a final concentration of 2.5 mM.

**Inducing MinDE detachment with sodium orthovanadate.** Na$_3$VO$_4$ stock solution was prepared from powder (Merck, Darmstadt, Germany) at a concentration of 200 mM. The solution was adjusted to pH 10 and heated alternately until the solution remained clear and colorless. MinDE self-organization in the presence of streptavidin was set up as described above. Self-organization of MinDE in the presence of streptavidin was imaged using definite focus. After several images, the scan head of the microscope was lifted to add Na$_3$VO$_4$ to the reaction at a final concentration of 2.5 mM. The opening of the scan head during the addition of Na$_3$VO$_4$ resulted in black images.

**PDMS microcompartment preparation.** Positive resist master molds of about 8 µm thickness were produced on a 4 inch silicon wafer (University Wafer) using ma-P 1275 (Microresist technology GmbH, Berlin, Germany) and a chrome mask (Compugraphics Jena GmbH, Jena, Germany), according to the manufacturer's data sheet and developed in ma-D531 (Microresist technology GmbH)[10]. We then spin-coated 200 µl of 1:20 Cytop CTL-809M in CTsolv.100E (both from Asahi Glass Co. Ltd., Japan) onto the master to ease later PDMS replica-molding. For this the Cytop dilution was directly pipetted onto the featured sections and spin-coated at 3000 rpm for 1 min, using a 500 rpm/s ramp. The wafer was then hard-baked for 30 min at 453 K on a hot plate to covalently anchor the coating, before being allowed to slowly cool down to room temperature by turning off the hot plate.

PDMS base and curing agent (Sylgard 184, Dow Corning Corporation, Michigan, USA) were mixed at a ratio of 10:1 in an ARE-250 mixer (Thinky Corporation, Tokyo, Japan). A drop of about 1–2 µl of PDMS was carefully placed on the master. Then a coverslide (thickness #1) was dropped on top and gently pressed down to squeeze the PDMS into a thin film. The wafer was then placed into an oven at 75 °C for at least one hour. Using a razor blade the coverslides with attached PDMS were removed from the Si wafer and a plastic sample chamber was glued onto the PDMS-covered slide. Directly before preparation of supported lipid bilayers the PDMS covered slide was placed into an oxygen plasma cleaner (Zepto, Diener electronic) and cleaned (1 min, 30% power, 0.3 mbar).

**Self-organization assay in PDMS microcompartments.** Self-organization assays in microcompartments were performed essentially as described earlier[10,66]. The self-organization assay was set up in 200 µl reaction buffer with 2.5 mM ATP, 1 µM MinD and 2 or 3 µM MinE. 0.5 µM mCh-MTS(BsD) was used for experiments with peripheral membrane proteins. In the case of lipid-anchored streptavidin membranes were prepared as described in streptavidin-bound membranes. After regular MinDE wave patterns formed on the surface of the PDMS, the volume of the buffer was lowered to the rim of the compartments by carefully removing buffer with a pipette. Hence, protein concentration inside the microcompartments are likely to be higher than original concentrations and are not comparable even between microcompartments in the same reaction chamber. A piece of sponge moistened in reaction buffer was plugged inside the reaction chamber to avoid drying of microcompartments and the chamber was sealed with a lid.

**Cholesterol-anchored and soluble P1 dsDNA fragments.** The DNA oligonucleotides FW_P1_30bp_sol, RV_P1_30bp_Al647 and FW_P1_30bp_chol were purchased from Eurofins Genomics (Ebersberg, Germany) and Sigma-Aldrich (St. Louis, USA), respectively. For DNA duplex formation oligonucleotides were dissolved in ddH$_2$O at 100 µM. The complimentary oligonucleotides were mixed in buffer (10 mM Tris-HCl pH 8.0, 1 mM EDTA, 50 mM NaCl) at a concentration of 10 µM each. They were annealed by slow cooling from 95 °C to room temperature in a heating block yielding 10 µM DNA duplexes. DNA duplex were added to self-organization assays at a final concentration of 10 or 100 nM, directly after MinDE addition. The sample was mixed by pipetting, lidded and incubated for more than 1 h before image acquisition.

**DNA anchored to lipid-anchored streptavidin.** The 300 and 2000 bp linear DNA fragments were generated by amplifying the first 300/2000 bp of lambda DNA (NEB, Ipswich, USA) by PCR using the forward primer BR215_Cy5_tetO_-lambda_fw and the reverse primers BR120_5′BiotinTEG_l300_rev and BR122_5′BiotinTEG_l2000rev (Sigma-Aldrich), respectively. The resulting PCR products were biotinylated and labeled with Cy5 on opposite ends. PCR products were purified and purity and labeling was assessed by gel electrophoresis. SLBs were generated as described under streptavidin-bound membranes using non-labeled streptavidin. After removal of surplus streptavidin from the reaction chamber, reaction buffer was added to a volume of about 50 µl and 6 pmol/2 pmol of the 300 bp/2000 bp long PCR product was added. DNA containing chambers were incubated for 2–3 h, then unbound DNA was removed by gently washing three times with a total of 600 µl reaction buffer. To start self-organization 1 µM MinD with 30% mol EGFP-MinD, 1 µM MinE, and 2.5 mM ATP in a total of 200 µl reaction buffer were added.

**Microscopy.** All images unless otherwise mentioned were taken on a Zeiss LSM780 confocal laser scanning microscope using a Zeiss C-Apochromat 40x/1.20 water-immersion objective (Carl Zeiss AG, Oberkochen, Germany). Longer time-series were acquired using the built in definite focus system. All two-color images were acquired with alternating illumination to avoid cross-talk. EGFP-MinD was excited

using the 488 nm Argon laser, mCh-MTS constructs using the 561 nm DPSS laser and streptavidin-Alexa647 or Cy5-DNA using the 633 nm He–Ne laser. Images were typically recorded with a pinhole size of 1 Airy unit, 512 × 512 pixel resolution, and a scan rate of 1.58 µs per pixel. Time-series for EGFP-MinD and mCh-MTS constructs were acquired with ~4 s intervals, EGFP-MinD and streptavidin-Alexa647 with ~5 s intervals.

Images in total internal reflection fluorescence (TIRF) microscopy were acquired on a custom-built TIRF microscope[69] using a NIKON SR Apo TIRF 100x/1.49 oil-immersion objective, constructed around a Nikon Ti-S microscope body (both Nikon GmbH, Düsseldorf, Germany). Two laser lines (490 nm (Cobolt Calypso, 50 mW nominal) and 640 nm (Cobolt 06-MLD, 140 mW nominal, both Cobolt AB, Solna, Sweden)) were controlled in power and timing (AOTF, Gooch&Housego TF-525-250, Illminster, UK) and spatially filtered (kineFLEX-P-3-S-405.640-0.7-FCS-P0, Qioptiq, Hamble, UK). The beam was further collimated, expanded (3×) and focused on the objective's back aperture by standard achromatic doublet lenses. The TIRF angle was controlled by precise parallel offset of the excitation beam (Q545, PI, Karlsruhe, Germany). For detection, two channels were separated by a dichroic mirror (Chroma T555lpxr-UF1), bandpass filtered (Chroma ET525/50m and ET670/30m, all Chroma Technology Cooperation, Bellow Falls, VT) and re-positioned on two halves of the EMCCD camera (Andor iXon Ulta 897). Images were recorded with Andor Solis (Ver. 4.28, both Andor Technologies, Belfast, UK).

**Image analysis**. All images were processed using Fiji[70] (version v1.51q) or Matlab (R2016a, The MathWorks, Natick, USA). Brightness or contrast adjustments of displayed images were applied homogenously.

**Analysis of mean fluorescence intensities**. Dual color time-series or tile scans were imported into Fiji and split into two separate image stacks. The EGFP-MinD stack was used to segment the MinDE waves in the images. To this end we used a custom-written ImageJ macro where the image from the EGFP-MinD channel was filtered using a median filter with radius 3–6 pixels, subsequently a "Pseudo-flat field correction" (BioVoxxel macro, Jan Brocher) with radius 75 pixels was applied to remove unequal illumination. The resulting image was thresholded using the Huang method or in the case of experiments with FtsZ-YFP-MTS or MinDE titration with the Li method, to generate the binary mask of the MinDE wave. This mask was also inverted to generate the complimentary mask.

The original non-modified images from the two spectral channels and the two complementary binary masks were imported into Matlab (R2016a, The MathWorks, Natick, USA) and analyzed using a custom-written Matlab code. The average fluorescence intensity in the mCherry/FtsZ-YFP-MTS ($I^{mCh-MTS}/I^{FtsZ-YFP-MTS}$) and EGFP-MinD/mRuby3-MinD ($I^{EGFP-MinD}/I^{mRuby3-MinD}$) spectral channel was obtained by pooling the means of individual images from one independent experiment. To obtain the average fluorescence intensity in the MinDE minima $\left(I^{mCh-MTS}_{min(MinD)}/I^{EGFP-MinD}_{min(MinD)}\right)$ and maxima ($I^{mCh-MTS}_{max(MinD)}/I^{EGFP-MinD}_{max(MinD)}$) the binary masks were multiplied with the original images of the respective spectral channels, all zero values were removed and the mean was taken. All means from one independent experiment and condition were pooled together. All fluorescence intensity values from one experimental set were normalized to the fluorescence intensity values obtained for His-mCherry for experiments with mCh-MTS and to a fluorescent standard for experiments with FtsZ-YFP-MTS. The contrast of the resulting protein waves was calculated for every individual image as the difference between the average intensity in the MinDE minima and MinDE maxima ($I^{protein}_{min(MinD)} - I^{protein}_{max(MinD)}$) divided by the average intensity in the MinDE maxima ($I^{protein}_{max(MinD)}$).

**Image preprocessing of FtsZ-YFP-MTS kymographs**. Image stacks were blurred (Gaussian blur) using Fiji. Afterwards every image was divided by its mean. This processed stack was used to produce kymographs shown in Fig. 5b.

**Analysis of fluorescence profiles in microcompartments**. Time-series from microcompartments were averaged in Fiji (version v1.51q) and the resulting average intensity was plotted over the full compartment and exported as csv file. Furthermore kymographs of every individual microcompartment were generated and used to assess MinDE oscillations. Microcompartments not showing MinDE oscillations were removed from further analysis.

To analyze the temporal averages of the spatial protein distributions in the microcompartments, we projected the fluorescence signal for each compartment on its elongated axis using Fiji. The obtained profiles (examples for EGFP-MinD (blue) and streptavidin (red) in Supplementary Fig. 14) were analyzed using a home-written MATLAB code (R2016a, The MathWorks). In a first step, the edge of the microcompartments was located along the MinD profile (blue line in Supplementary Fig. 14) based on the increase of EGFP signal and the concomitant change of the first spatial derivative. The initial profiles were clipped accordingly. Subsequently, the two local maxima of the MinD profile were located in a two step procedure: First, their location was roughly estimated based on a polynomial fit of fourth order. Second, a 40 pixel region of interest was selected around these estimated positions and the corresponding section of the profile was fitted with a

quadratic function to locate the maximum more precisely. The positions of the located maxima are defining the edges of a unit box, onto which the profiles of both spectral channels were projected. We were seeking an easy way to classify the profiles in this unit box and therefore decided to fit the profiles with a quadratic function $f(x)=ax^2+b$ (black dashed lines in Supplementary Fig. 14a), where $2a$ represents the overall curvature and hence the steepness of the profile and $b$ accounts for the offset. As we projected the profiles onto a unit box of length 1, the depth of the profile and the curvature are identical, except for a constant prefactor, and are thus interchangeable terms. Homogenously distributed fluorescence corresponds to a curvature of $a \approx 0$. A spatial distribution with enrichment in the center of the compartment yields $a<0$, whereas proteins that are on average less likely to be found in the center will be classified with a curvature $a>0$. In this classification, the MinD profile has a curvature $a>0$.

## Data availability

Data supporting the findings of this manuscript are available from the corresponding author upon reasonable request. The custom-written code for the analysis of the time-averaged fluorescence profiles in microcompartments can be found on github (https://github.com/BeaRamm/intensity_profiles). All other code is available from the corresponding authors upon reasonable request.

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

## Acknowledgements

We thank MPIB Core Facility for assistance in protein purification, Stephan Uebel for peptide synthesis, Katharina Nakel and Michaela Schaper for assistance with cloning, Lei Kai for plasmid pCoofy1-mCherry, Martin Loose for plasmid pML60, Frank Siedler for help with Si wafer fabrication, Jan Brocher from BioVoxxel for help with generating the Fiji macro and Ana Raso for establishing the FtsA purification protocol. Further we thank Kristina Ganzinger, Henri Franquelim, and Simon Kretschmer for helpful discussions, and Kristina Ganzinger and Hiromune Eto for comments on the manuscript. B. R. and D.G. are supported by a DFG fellowship through the Graduate School of Quantitative Biosciences Munich (QBM). B.R and P.S. acknowledge funding through the DFG Collaborative Research Centre "Spatiotemporal dynamics of bacterial cells" (TRR 174/2017). P.S. acknowledges the support of the research network MaxSynBio via a joint funding initiative of the German Federal Ministry of Education and Research (BMBF) and the Max Planck Society. J.M. is grateful for financial support from the excellence cluster Nanosystems Initiative Munich. J.M., P.B. and P.G. acknowledge support from the International Max Planck Research School for Molecular Life Sciences. P.G. acknowledges support from GRK 2062—Molecular Principles of Synthetic Biology. MH

acknowledges support from the Joachim Herz Foundation through an Add-on Fellowship. We acknowledge support from the Center for NanoScience Munich.

## Author contributions

B.R. designed and performed experiments. B.R. and P.S. conceived the study and wrote the manuscript. B.R. and J.M. analyzed the data. P.B. performed TIRF experiments. P.G. designed and purified proteins. M.H. manufactured Si wafer for microcompartments. D.G. purified FtsZ-YFP-MTS. All authors discussed and interpreted results and revised the manuscript.

## Additional information

**Competing interests:** The authors declare no competing interests.

