## [Peer Review File · Nature Communications]

Reviewers' comments:

Reviewer #1 (Remarks to the Author):

Report on Ramm et al (NCOMMS-18-03549-T)

This paper describes the observation of the spatiotemporal organization of membrane associated proteins by a reconstituted Min system in vitro. The authors show that the MinDE protein system is capable of regulating a larger set of membrane proteins than previously known. They propose a mechanism according to which a cooperative membrane self-organization of Min patterns on supported lipid bilayers is able to out-compete weakly membrane associated proteins, leading to their localization in the Min protein minima. Proteins that cannot be easily dissociated from membrane due to stronger interactions with the membrane are moved along the membrane due to diffusion barriers induced by the Min system. A similar effect was observed for membrane-anchored DNA molecules. As an extrapolation of these in vitro results, the authors propose, that Min system oscillations in *E. coli* may use a similar mechanism of protein relocation on the membrane to contribute to chromosome segregation and to mid-cell positioning of some membrane proteins involved in cell division like FtsA/ZipA.

This in vitro study reports an interesting property of the reconstituted Min system that was not reported before. The finding is not entirely unexpected, as it is likely that proteins with a strong membrane affinity and self-organization properties will exclude other membrane components. Overall, the study is of interest, although I think that it may be more fitting for a specialized journal than for Nature Communications which aims to publish the top papers in the field.

Yet, there are major problems in the paper. Most specifically, the proposed mechanism of protein localization regulation in *E. coli* is interesting but it has two major weak points which are not addressed in the current work:

1) The authors provide evidence that Min proteins can move proteins which are functionally unrelated to Min proteins. If the Min system does indeed play the suggested role in *E. coli*, all inner membrane (almost 600 different membrane-embedded and membrane-associated proteins!), or at least all membrane associated proteins (280 proteins; H-L Lee et al., 2016) would show an oscillatory behavior. But this does not seem to be the case, and would also not be reasonable from the point of view of cellular biochemistry. The authors seem to ignore this point.

2) The spatiotemporal positioning of proteins on the membrane by the in vitro Min system seems to be rather an artifact caused by specific properties of the Min system reconstituted in the specific environment. Min proteins reconstituted in open confinements like the SLBs in the reaction chamber and microwells prepared in a way presented in this paper accumulate on the membrane to maximal densities around 1.6×10^4 MinD/ m^2 which are more than an order of magnitude higher than the biologically relevant densities in *E. coli* cells (800 MinD/ m^2 assuming MinD being localized at one pole of an *E. coli* cell with an average total surface area of 4 m^2). This dramatically changes the concept. While high Min membrane protein concentrations may form a diffusion barrier, the situation in living cells rather looks drastically different, and Min system likely does not function this way.

Other remarks:

- All the presented experiments are missing basic characterizations of the systems used. Measurements of the surface protein concentration, the membrane coverage (MinCD as well as the co-regulated components) as well as e.g. diffusion constants of certain constructs would give the reader basic information that is important to understand what happens in these experiments.

- In all figures, the authors present pictures from two channels: MinD and an other protein or DNA. In these cases, it is advised to also show composite pictures. While for mCherry constructs it is relatively easy to visually superpose both channels, in streptavidine and DNA experiments, the location of specific gradients is much more complex.

- In fig 2b, the control experiment of His-mCH captured at 0-10000 threshold shows a basal intensity equal to areas where the MTS(1xMreB)-mCh construct is enriched at the membrane. Does this indicate some protein interaction with the surface? The authors should comment on that.

- In fig 3a one can observe that the gradients of streptavidine become less pronounced over time. Is this a typical observation? Does it mean that over time these gradients undergo equalization?

- The experiments presented here raise a question regarding previously published observations (also published by the Schwille group: Arumugamet al., 2014, PNAS) where MinDE patterns reconstituted on the membrane (in the absence of MinC) did not lead to FtsZ-mts construct repositioning. The authors should explain this. If the reason would be the polymerization state of FtsZ, then how does it relate to the experiments with FtsA (Supplementary Figure 5) which is also known to form oligomers?

- In Supplementary Figure 5, the authors show what I think is one of the most important and interesting experiment in the entire manuscript, namely the positioning of FtsA by MinDE waves. In my opinion, this figure should be extended with an exhaustive analysis and likely be contained in the main manuscript.

- In Supplementary figure 1d,e,f, the authors calculate and compare average MinD intensities. It is hard to follow the normalization that then authors did. Even though the difference between intensities in Min wave minima and maxima are very pronounced for these experiments, the denoted intensities don't seem to differ between panels d,e,f in the presented plots. It is therefore not clear how the authors obtained these data.

- The Materials and methods section contains mistakes (e.g. in line 567 and 569, the authors use buffer at pH 80, numerical values are notoriously not spaced from units, which should be the standard according to SI Unit rules). This section should be carefully corrected.

- The FtsA purification protocol (line 550) seems to be missing any lysogenic agent or procedure (e.g. lysozyme and sonication). The procedure is also missing protein purity control.

- The manuscript is missing information regarding the temperature at which the experiments were performed.

- Why did the authors not focus more on generic E coli membrane proteins? I found it somewhat odd that in Fig.1 the authors study a B. subtilis protein as their prime/first example of a different protein that is modulated by the E. coli MinDE system.

- The Materials and methods section lacks important information about the preparation of experiments with mCherry constructs and FtsA. For example, were these proteins added to the reaction volume or was the membrane first pre-incubated with these proteins and MinDE proteins were added later.

- I dislike 'cheap sentences' like the last line of the abstract "This previously underestimated capacity of reaction diffusion systems to actively transport membrane proteins may be key to their function in bacteria and eukaryotes.". I suggest to be more modest and tone down the phrasing.

Reviewer #2 (Remarks to the Author):

This is an excellent study with novel and important discoveries. Previous studies from the Schwillie lab have shown that waves of MinCDE on slb generate anticorrelated waves of FtsZ-mts. This was attributed to the effect of MinC in causing disassembly of FtsZ protofilaments and/or bundles. The present study shows that MinDE waves, without the inhibitor MinC, generate anticorrelated waves of non-specific proteins tethered to the membrane by mts amphipathic helices. A rather different wave is achieved by proteins linked by TM insertions, which can diffuse in the membrane but are not reversibly dissociating. This suggests that at least some of the inhibition of FtsZ by MinCDE may be due to steric inhibition on the membrane by MinDE, independent of MinC. Of course it would be interesting to see directly that MinDE can set up waves of FtsZ-mts and FtsZ-FtsA, but that can be a future study. The experiments presented here are convincing and well described. I recommend publication following attention to a few very minor concerns.

Biotinyl-CAP-PE needs a description. Fig. 3b shows two lipids inserted into the bilayer quite distant from each other. The Avanti web page shows a single lipid.

Fig. 5d was not clear to me. In particular I did not understand "unit box" and the units -0.5 to +0.5.

Reviewer #3 (Remarks to the Author):

In this paper the authors further explore features of the Min system in vitro and show that peripheral membrane proteins counter oscillate with MinD. The pictures and movies are very convincing. This is shown in both their 2D and 3D systems and it is shown that it is fairly independent of the membrane targeting sequence they use. They also show that for a protein anchored to the membrane the behavior is quite different than that of the peripheral membrane protein. They also turn their attention to testing possible interaction of the Min system with DNA. This was stimulated by a report that MinD bound DNA. They find no evidence of MinD binding DNA but the Min pattern formation causes counteroscillations of the DNA in their system. The results lead to speculation about the possible influence of the Min system on proteins and DNA segregation that have not been directly linked to the Min system. In my opinion this is quite dangerous as the in vitro and in vivo systems are dramatically different as indicated below in my major concern.

Main concern.

The results are very clear in the in vitro system. My major concern is what it means. In the discussion the authors extrapolate to the in vivo situation. However, there is no data, only speculation. In this paper they estimate the MinD in the peak of the wave to be 16,000 per square micron. This is consistent with their in vitro explanation that MinD almost saturates the membrane and therefore excludes peripheral membrane proteins and acts as a diffusion barrier to proteins attached to the lipids. However, in vivo the estimate for MinD is about 200 per square micron (ref 12). Thus, the in vivo and in vitro systems are quite different. I don't understand why this wasn't addressed.

The authors also looked at the potential interaction of MinD with DNA and found none. I do not find this surprising as the one report appeared badly flawed. The MinD mutant reported to affect MinD binding actually affected membrane binding.

In an earlier paper this group showed that Min counter oscillated with FtsZ. In that study the counteroscillation depended upon MinC; MinD/MinD alone could not do it. What is the difference with proteins here. Is it that FtsZ is polymerized?

Reviewer #4 (Remarks to the Author):

Ramm, et al., observed that the MinDE system can spatiotemporally regulate the distribution of model membrane-associated and membrane-anchored proteins *in vitro*. The authors propose that this is a novel/newly described feature of the MinDE system to systematically regulate the position of membrane-bound proteins by causing redistribution in the membrane. The authors also speculate that this mechanism may be important for regulating the midcell localization of cell division components, other membrane proteins and also modulating chromosome segregation *in vivo*. The data are presented in a very organized and well-communicated manner. The techniques and observations are interesting and compelling, but it is difficult to extend these observations to the *in vivo* physiological system. In an *in vivo* model, one could then test the relevant impact of tethering large complexes, or DNA, or other redistribution and compartmentalization scenarios including multispinning transmembrane domains.

There are several concerns regarding experimental constraints/parameters that impact the generalizable nature and relevance of the observations. The authors are asked to address the following comments.

1. Most of the experiments are performed at 1 μM MinDE in a chamber, where the surface waves of MinD represent regions of membrane-bound protein at high local concentration of MinD. The width of the MinD zone (i.e., in Fig. 1a) captured in the surface wave is 100 μm . The amount of MinD required to produce this zone seems disproportionately higher than the available MinD in a typical cell. If MinD is functionally capable of displacing surface associated proteins in an *in vivo* system, such as an *E. coli* cell, it must be densely populating the inner surface of the membrane. Is there enough MinD in a given cell to populate a membrane surface zone such that it behaves as in the system demonstrated *in vitro*. If there is, then is the zone expected to be smaller than the 100 μm zone depicted in the figure (1a)? And if substantially smaller, would it produce the same effects? What is the surface density required to displace? If the surface density of the min wave is increased, it should then be more productive at displacing/reorganizing. The cell-shaped microcompartments, if physiologically analogous or relevant, should also be designed to recapitulate MinDE cellular concentration, lipid content, membrane fluidity, etc.
2. The surface wave zones exhibit varying widths across the experiments. For example, zones depicted in Fig. 2b range from ~ 20 μm to ~ 60 μm . Are the zones traveling at different rates? What is the mean deviation of zone widths and deviation of the rate of the surface waves? What is responsible for this variation? Is this performed under steady-state conditions in which ATP is not limiting and ADP is not accumulating?
3. In addition to the strength of the interaction between the membrane-targeting region of the displaced protein and the membrane, a major impacting factor on redistribution of displaced protein should be membrane fluidity. If lateral diffusion in the membrane is affected, displacement, particularly by a permanently attached membrane protein, should also be affected. What are the effects of modifying membrane fluidity, either by altering lipid content or temperature, on redistribution of displaced protein. The permanent-attachment mimic used a nonpolar/lipid anchoring group that was fused to a protein. However, this would likely be far more diffusible than a full transmembrane domain. If diffusion is limited by reduced fluidity or a bulkier transmembrane(s), would the hindrance be sufficient to induce breakdown of the MinD surface wave propagation?
4. Does an mCherry with an MTS from *E. coli* MinD and/or MinE also become excluded? These fusions should have similar affinities for the membrane at MinD surface waves, excluding surface dimerization effects.

5. FtsA was suggested to behave as a surface associated protein (Fig. S5). However, FtsA has been reported to oligomerize on the membrane, promote membrane distortion, hydrolyze ATP rapidly and recruit FtsZ polymers to a membrane (Loose, 2014; Krupka, 2017; Conti, 2018). These activities likely complicate the predicted behavior and could explain what FtsA counteroscillations have not been observed in vivo.

6. The authors found that MinDE propagation on lipid bilayers can spatially regulate the membrane distribution of mCherry fusion proteins containing amphipathic helices and modeled transmembrane domains. Based on these observations, the authors propose a novel function of the MinDE system to spatiotemporally regulate membrane-associated and membrane-anchored proteins. Specifically, the authors propose that MinDE-driven counter oscillations of ZipA, which were previously reported (Bisicchia, 2013) leads to ZipA enrichment at midcell. By expressing Gfp-tagged ZipA in live cells, ZipA localization in wildtype cells can be monitored and compared to a minE deletion strain. If the MinDE system is important for septal localization of ZipA, then disrupting the MinDE system should lead to aberrant ZipA localization. If the ZipA transmembrane domain is replaced with an amphipathic helix, does the behavior or patterning change in vivo? This, and similarly designed experiments to test the predicted mechanism in vivo, as well as min-dependent localization of ZapB and MatP, which contacts the chromosome, would add physiological context to the study. In addition, distribution of fluorescent proteins in vivo in the min+ and min- strain could also be monitored for other proteins that are recruited to the membrane but do not participate in cell division, which would indicate the general applicability of the positioning or membrane-partitioning system.

7. The authors observe that the spatial distribution of mCherry-MTS constructs is regulated by MinDE propagation in vitro. The authors confirmed that soluble His-mCherry is not spatially regulated by MinDE. However, if the concentration of His-mCherry were higher, would this remain true? Does molecular crowding affect the min surface wave or mass transfer?

8. Cells deleted for minC are associated with chromosome segregation defects in vivo (Akerland, et al., 2002), suggesting that MinC may contribute to the phenotype. It is possible that reduced minDE expression in minC mutants accounts for chromosome segregation defects, and that overexpression of minDE would promote proper chromosome segregation. A second possibility is that MinC alters surface wave propagation and reduces efficiency of directing chromosome segregation. The authors suggest that MinD could assemble into higher order structures (line 409). In fact, MinD is reported to form copolymers with MinC (Ghosal, et al., 2014; Conti, 2015).

9. The authors report that DNA bound to membrane tethers is spatially regulated by MinDE propagation on lipid bilayers independently of a direct protein-DNA interaction, and propose that the MinDE system can affect chromosome segregation in vivo by regulating the distribution of membrane-associated DNA binding proteins. FtsK is a membrane-anchored protein that regulates E. coli chromosome segregation and is localized to the septum (Yu, et al., 1998). If the MinDE system spatiotemporally regulates membrane-bound DNA binding proteins, then septal localization of FtsK-Gfp may be perturbed in min- cells. The authors are asked to examine the affect of disrupting MinDE on localization of membrane-anchored DNA-binding proteins in live cells.

We thank the reviewers for the constructive comments on the manuscript and would like to address the two main concerns brought forward up front before responding to the individual points.

1. Differences between *in vivo* and *in vitro* MinDE self-organization

The major concern of the reviewers was the comparability between the experiments *in vitro* and the potential situation *in vivo*.

In vivo the amount of MinD and MinE molecules was determined to be between 2000-3000 molecules per cell with an average membrane area of $6 \mu\text{m}^2$. Assuming that all MinD proteins bind to the membrane and localize only to one pole in the cell at a time, the estimated MinD density on the membrane would be about $1 \times 10^3 \mu\text{m}^{-2}$ (New Supplementary Note 1) (Shih, Y. L., Fu, X., King, G. F., Le, T. & Rothfield, L. EMBO J. 21, 3347–3357 (2002); de Boer, P. A., Crossley, R. E., Hand, A. R. & Rothfield, L. I. EMBO J. 10, 4371–4380 (1991).

In vitro our group and others have measured MinD densities of about $5 - 16 \times 10^3 \mu\text{m}^{-2}$ (Vecchiarelli, A. G., Li, M., Mizuuchi, M. & Mizuuchi, K. Mol. Microbiol. 93, 453–463 (2014); Loose, M., Fischer-Friedrich, E., Herold, C., Kruse, K. & Schwillie, P. Nat. Struct. Mol. Biol. 18, 577–583 (2011)).

We have proposed that the mechanism behind the generic spatiotemporal positioning by MinDE is that MinDE form a propagating diffusion barrier that outcompetes other proteins during membrane attachment and drives the directed motion of laterally diffusing molecules.

Given the about one order of magnitude higher densities observed *in vitro* compared to the estimated density *in vivo*, the reviewers suggested that while this mechanism might be at work *in vitro*, the situation is different *in vivo* and that conditions *in vivo* might not support the spatiotemporal positioning by MinDE.

To address this issue we have performed a variety of additional experiments.

We performed a titration series of MinDE (new Supplementary Figure 5) showing that the positioning of the mCh-MTS construct with the highest membrane affinity, MTS(2xMreB)-mCh, can be observed for all tested MinD/MinE ratios (10 – 0.1) (Supplementary Fig. 5 a,b), at the lowest equimolar MinDE concentration that still supported self-organization in our *in vitro* assay (MinDE = 0.4 μM) (Supplementary Fig. 5c-e), and under all mCh-MTS/MinDE ratios tested, as high as 30 and as low as 0.1 (Supplementary Fig. 6). We further quantified MinD and MTS(2xMreB)-mCh densities on the membranes using a combination of Fluorescence Correlation Spectroscopy (FCS) and imaging (see new Supplementary Figure 11 for details on the calibration). For the standard concentrations used in our study (1 μM MinD, 1 μM MinE) we obtain a MinD density on the membrane of around $1.3 \times 10^4 \mu\text{m}^{-2}$ which is similar to the previously reported values (new Supplementary Fig. 13). However, for the lowest MinDE concentrations that still produced patterns in our *in vitro* setup (0.4 and 0.5 μM) and are also able to spatiotemporally regulate model peripheral membrane proteins, we measure concentrations of $1.8 \times 10^3 \mu\text{m}^{-2}$ and $3.3 \times 10^3 \mu\text{m}^{-2}$, that are on the same order of magnitude as the estimated *in vivo* densities (new Supplementary Fig.13, new Supplementary Note 1).

Of course the cellular membrane is different from our model membranes, as already a large fraction (~60%) of the total cellular membrane surface is occupied by transmembrane proteins (Devaux, P.F., and Seigneuret, M. *Biochim Biophys Acta* 822, 63–125 (1985)). However, MinDE patterns are able to couple across membrane gaps (Schweizer, J. et al. Proc. Natl. Acad. Sci. 109, 15283–15288 (2012)) or immobile structures that cannot be laterally moved by MinDE *in vitro* (see experiments with FtsZ-YFP-MTS, new Fig. 5 and Supplementary Fig. 16) and hence are likely to also do so *in vivo*. Thus, the accessible membrane area for MinDE binding *in vivo* might well be reduced, increasing local protein densities.

Further, we would like to point out that MinDE are able to spatiotemporally regulate lipid-anchored streptavidin that crowds the membrane quite substantially. The density of streptavidin can be calculated from the amount of biotinylated lipids and the streptavidin-biotin valency (Dubacheva, G.

V. *et al.* *J. Am. Chem. Soc.* **139**, 4157–4167 (2017) to 6600 molecules/ μm^2 and we obtained similar values when we measured the streptavidin densities as a control for our FCS-based image calibration. Assuming a streptavidin size of 5 nm by 5 nm, streptavidin would cover about 17% of the total membrane area. Hence, even under crowding conditions on the membrane MinDE are able to spatiotemporally regulate the crowder. This also explains why MinDE pattern formation is significantly influenced by lipid-anchored streptavidin, reducing its density and wavelength (see Fig. 3d).

Furthermore, we show that MinDE are able to regulate three additional model peripheral membrane proteins that contain one or two copies of the native *E. coli* MinD MTS (new Supplementary Fig.7). The mCh-MTS constructs harboring two MinD MTS should have a similar membrane affinity as the alleged MinD species on the membrane, a dimer. Hence, this results questions the commonly accepted MinDE self-organization mechanism, suggesting instead that MinD membrane binding includes higher order recruitment or oligomerization. This again supports the hypothesis that MinDE can act as a propagating diffusion barrier when individual MinD dimers interact on a higher order on the membrane.

We now highlight the differences between *in vitro* and *in vivo* conditions in the updated discussion and also point out that MinDE have been shown to alter the physical properties of membranes, which could be an alternative mechanism of action.

Our results obtained *in vitro* are generic and show that MinDE are able to regulate a variety of membrane-attached molecules. This however does not imply that all membrane bound molecules in *E. coli* are regulated by MinDE. For example, transmembrane proteins that are not freely diffusing in the membrane because they are stably anchored to the cell wall or larger filamentous structures such as MreB filaments will likely act as static obstacles and will not be regulated by MinDE (see Supplementary Fig. 15 showing that MinDE cannot laterally move FtsZ-YFP-MTS, or Supplementary Movie 14, showing that streptavidin crystals on the membrane also remain static). Transmembrane proteins with no or small cytosolic domain would neither be regulated. Further, proteins that favor a certain lipid composition or binding partner might be subject to other, stronger spatial cues. We have included this point into the updated discussion.

Furthermore, we provide several other new experiments showing that the spatiotemporal regulation by MinDE is generic *in vitro*.

We now show that while the spatiotemporal regulation is independent of MinC, it is also occurring in the presence of MinC, which is of course an integral part of the MinCDE system in *E. coli* (new Supplementary Fig. 10).

We have re-purified and relabeled FtsA using sortase-based labeling introduced by Loose *et al.* (Loose, M. & Mitchison, T. J. *Nat. Cell Biol.* **16**, 38–46 (2014)). Using this strategy we obtained a protein less prone to aggregation and with higher labeling ratio. Like the maleimide-labeled FtsA this protein is also regulated by MinDE (see new Supplementary Fig.15), but contrary to the old experiments shows a behavior that is similar to lipid-anchored streptavidin, suggesting that indeed FtsA oligomerizes *in vitro* and hence does not behave like mCh-MTS constructs.

We also show that MinDE is able to spatiotemporally regulate the chimeric protein FtsZ-YFP-MTS (see details below).

Of course, all these *in vitro* results do not ultimately clarify whether this non-specific spatiotemporal regulation is occurring *in vivo* and which specific proteins or DNA would be regulated to what extent. We argue that a detailed *in vivo* study is needed to answer this question which is not in the scope of this study. Hence, we have toned down our claims on the occurrence of the mechanism *in vivo* in the updated discussion.

However, we hope that this detailed *in vitro* study will motivate future *in vivo* studies. Furthermore, this in-depth characterization will enable us and others to use the MinDE system to transport and position arbitrary molecules in artificial cells in the future, an application we had not included into the manuscript previously, but rather emphasized of a potential physiological role of this mechanism. We have now included this important perspective.

2. Regulation of FtsZ-YFP-MTS

Another main concern raised by several reviewers was that we had previously shown that the chimeric protein FtsZ-YFP-MTS was not regulated by MinDE alone, but only if MinC was supplied (compare to Arumugam, S., Petrašek, Z. & Schwille, P. Proc. Natl. Acad. Sci. U. S. A. 111, E1192–E1200 (2014) and Zieske, K. & Schwille, P. Elife 3, e03949 (2014)).

To address this concern we have revisited the previous experiments and included them in the manuscript as new Fig. 5 and Supplementary Fig. 15).

These two previous studies used the chimeric FtsZ-YFP-MTS under similar conditions, namely high free Mg^{2+} in the buffer, that led to large FtsZ filament bundles on the membrane.

Our group has recently shown that FtsZ-YFP-MTS is also able to form dynamic ring-like structures, similar to experiments with co-reconstituted FtsA and FtsZ, when the GTP concentration is increased to 4 mM GTP, reducing the free Mg^{2+} concentration to ~ 1 mM $MgCl_2$ in the assay (Ramirez-Diaz, D. A. *et al. PLoS Biol.* **16**, e2004845. (2018)). This change from filaments to dynamic rings was accompanied by a shorter membrane residence time of FtsZ-YFP-MTS monomers and a decreased total protein density on the membrane.

We have therefore conducted an experiment with MinDE and 0.5 μ M FtsZ-YFP-MTS under the two different free Mg^{2+} concentrations: high free Mg^{2+} (~ 5 mM), where FtsZ-YFP-MTS forms large bundles, and low free Mg^{2+} (~ 1 mM) where FtsZ-YFP-MTS forms small rotating rings, with and without 0.05 μ M MinC.

Under conditions similar to our previous reports (high free Mg^{2+}), the spatiotemporal regulation of FtsZ-YFP-MTS filaments is very hard to detect. The difference in brightness cannot be seen in individual images by eye, but only when looking at an image sequence (see new Supplementary Movie 9). For showing the spatiotemporal regulation in kymographs the images needed to be preprocessed (see new Fig. 5b). In contrast, for the conditions under which FtsZ-YFP-MTS forms dynamic rings (low free Mg^{2+} concentrations), the spatiotemporal regulation by MinDE only is clearly discernible. When MinC is supplied under either of these conditions, FtsZ-YFP-MTS is mostly disassembled and only bound to the membrane in the MinCDE minima, leading to a very strong regulation.

Thus, in the past the weak spatiotemporal regulation of FtsZ-YFP-MTS when forming bundles (high free Mg^{2+}) was masked when images of the two spectral channels were taken at the same time, rather than alternatingly as in this study.

These new experiments clarify a possible confusion that might have arisen for readers that are familiar with our group's older publications by showing all conditions side by side.

More importantly they show that even treadmilling proteins such as FtsZ can be spatiotemporally regulated by MinDE alone, although MinC drastically increases the efficiency of the regulation. The MinDE-dependent regulation by a propagating diffusion barrier can thus be seen as an archetypal physicochemical mechanism. MinC, on the other hand, augments it while conferring protein specificity.

Furthermore, looking at the filaments and rings in the presence of MinDE with higher magnification reveals that these rings or filaments are too strongly attached in order to diffuse on the membrane. Hence, MinDE cannot laterally move FtsZ-YFP-MTS as in the case of lipid-anchored streptavidin. The weak spatiotemporal regulation of the treadmilling FtsZ-YFP-MTS is more likely caused by MinDE regulating the membrane attachment of FtsZ-YFP-MTS monomers (new Supplementary Fig. 16,

Supplementary Movie 10). Hence, this experiment also explains partially why not all membrane-bound proteins in an *E. coli* cell would be regulated by such a generic mechanism (see answer above).

For the more specific comments by the reviewers, please see the point by point answer below.

Reviewers' comments:

Reviewer #1 (Remarks to the Author):

Report on Ramm et al (NCOMMS-18-03549-T)

This paper describes the observation of the spatiotemporal organization of membrane associated proteins by a reconstituted Min system in vitro. The authors show that the MinDE protein system is capable of regulating a larger set of membrane proteins than previously known. They propose a mechanism according to which a cooperative membrane self-organization of Min patterns on supported lipid bilayers is able to out-compete weakly membrane associated proteins, leading to their localization in the Min protein minima. Proteins that cannot be easily dissociated from membrane due to stronger interactions with the membrane are moved along the membrane due to diffusion barriers induced by the Min system. A similar effect was observed for membrane-anchored DNA molecules. As an extrapolation of these in vitro results, the authors propose, that Min system oscillations in *E. coli* may use a similar mechanism of protein relocation on the membrane to contribute to chromosome segregation and to mid-cell positioning of some membrane proteins involved in cell division like FtsA/ZipA.

This in vitro study reports an interesting property of the reconstituted Min system that was not reported before. The finding is not entirely unexpected, as it is likely that proteins with a strong membrane affinity and self-organization properties will exclude other membrane components. Overall, the study is of interest, although I think that it may be more fitting for a specialized journal than for Nature Communications which aims to publish the top papers in the field.

Yet, there are major problems in the paper. Most specifically, the proposed mechanism of protein localization regulation in *E. coli* is interesting but it has two major weak points which are not addressed in the current work:

We thank the reviewer for his/ feedback and hope to address his/her major points in the answer to all reviewers above and in the points below.

1) The authors provide evidence that Min proteins can move proteins which are functionally unrelated to Min proteins. If the Min system does indeed play the suggested role in *E. coli*, all inner membrane (almost 600 different membrane-embedded and membrane-associated proteins!), or at least all membrane associated proteins (280 proteins; H-L Lee et al., 2016) would show an oscillatory behavior. But this does not seem to be the case, and would also not be reasonable from the point of view of cellular biochemistry. The authors seem to ignore this point.

Please see the answer to all reviewers above.

2) The spatiotemporal positioning of proteins on the membrane by the in vitro Min system seems to be rather an artifact caused by specific properties of the Min system reconstituted in the specific environment. Min proteins reconstituted in open confinements like the SLBs in the reaction chamber and microwells prepared in a way presented in this paper accumulate on the membrane to maximal densities around 1.6×10^4 MinD/ m^2 which are more than an order of magnitude higher than the biologically relevant densities in *E. coli* cells (800 MinD/ m^2 assuming MinD being localized at one pole of an *E. coli* cell with an average total surface area of 4 m^2). This dramatically changes the

concept. While high Min membrane protein concentrations may form a diffusion barrier, the situation in living cells rather looks drastically different, and Min system likely does not function this way.

Please see the answer to all reviewers above.

Other remarks:

- All the presented experiments are missing basic characterizations of the systems used. Measurements of the surface protein concentration, the membrane coverage (MinCD as well as the co-regulated components) as well as e.g. diffusion constants of certain constructs would give the reader basic information that is important to understand what happens in these experiments.

Please see the answer to all reviewers above. We have added a characterization of surface protein densities to the manuscript (new Supplementary Fig. 13).

- In all figures, the authors present pictures from two channels: MinD and an other protein or DNA. In these cases, it is advised to also show composite pictures. While for mCherry constructs it is relatively easy to visually superpose both channels, in streptavidine and DNA experiments, the location of specific gradients is much more complex.

We have attached composite images for all main figures that include Streptavidin or DNA and EGFP-MinD in the new Supplementary Figure 8. We have not included those pictures in the main text, as we do not think they add more information, but rather mask subtle details.

- In fig 2b, the control experiment of His-mCh captured at 0-10000 threshold shows a basal intensity equal to areas where the MTS(1xMreB)-mCh construct is enriched at the membrane. Does this indicate some protein interaction with the surface? The authors should comment on that.

We thank the reviewer for pointing out this discrepancy. We set out to determine the origin of this difference and re-measured all concentrations of mCh-MTS constructs using Bradford assay and also determined the amount of fluorescent mCherry by measuring absorption spectra. Even though all proteins have been expressed and purified using the same protocol, the fluorescent fraction varies quite strongly between the different proteins. This might be due to influences of the different amphipathic helices on the mCherry fluorophore or due to differences in maturation. We have therefore corrected the intensities in Figure 2d and Supplementary Fig. 3 (former Supplementary Fig.1) for the fraction of fluorescent mCherry. Furthermore, we acquired z-stacks of an assay containing the different mCh-MTS constructs and a fluorescently labeled membrane in the absence of MinDE (new Supplementary Fig. 1). While this is not an accurate measure of membrane binding the z-stacks qualitatively show that His-mCh does not bind to the membrane, whereas MTS(1xMreB)-mCh binds to the membrane, albeit very weakly.

- In fig 3a one can observe that the gradients of streptavidine become less pronounced over time. Is this a typical observation? Does it mean that over time these gradients undergo equalization?

During the spatiotemporal regulation of membrane-anchored streptavidin by MinDE two kind of intensity differences emerge. At the onset of the MinDE self-organization only short-ranged concentration differences arise as the streptavidin accumulates in the minima of the MinDE wave. Over time the large-scale gradients develop, where the streptavidin is depleted from spiral centers and accumulates where wave fronts collide. Due to the underlying larger gradients the small-scale gradients, namely the accumulation in the MinDE minima, depending on their position in the larger gradients are weaker. However, even in regions where the streptavidin is almost entirely depleted, one can still see that the protein accumulates in MinDE minima. Please see the kymograph in Figure 7a, and the heavily depleted regions shown in Figure 3d.

- The experiments presented here raise a question regarding previously published observations (also published by the Schwille group: Arumugamet al., 2014, PNAS) where MinDE patterns reconstituted on the membrane (in the absence of MinC) did not lead to FtsZ-mts construct repositioning. The authors should explain this. If the reason would be the polymerization state of FtsZ, then how does it relate to the experiments with FtsA (Supplementary Figure 5) which is also known to form oligomers?

Please see the answer to all reviewers above. We have added two new figures to clarify this point (new Fig 5, new Supplementary Fig. 16).

Regarding the point about FtsA, please see the comments below.

- In Supplementary Figure 5, the authors show what I think is one of the most important and interesting experiment in the entire manuscript, namely the positioning of FtsA by MinDE waves. In my opinion, this figure should be extended with an exhaustive analysis and likely be contained in the main manuscript.

We have re-purified and labeled FtsA according to a protocol by Loose *et al.* (Loose, M. & Mitchison, T. J. *Nat. Cell Biol.* **16**, 38–46 (2014)). Using this protocol we obtained a protein less prone to aggregation, with better labeling ratio and a seemingly higher membrane affinity, that allowed us to observe MinDE-dependent positioning of FtsA on the confocal microscope instead of TIRF microscopy. The general outcome of the experiments has not changed, in that a spatiotemporal regulation of FtsA by MinDE can be clearly observed. However, when comparing the kymographs of FtsA to the kymographs of the model peripheral membrane proteins (mCh-MTS) and the lipid-anchored streptavidin, FtsA seems to behave more similar to the lipid-anchored streptavidin, forming, albeit very weakly, large-scale gradients.

This behaviour could indeed be due to a higher order oligomerization state of FtsA on the membrane. We have changed the respective section in the manuscript, clearly stating that FtsA does not behave like a monomeric, peripheral membrane protein.

FtsA is known to be difficult to work with *in vitro*, and different oligomerization states and dynamics have been observed (Loose, M. & Mitchison, T. J. *Nat. Cell Biol.* **16**, 38–46 (2014)., Krupka, M. et al. *Nat. Commun.* **8**, 1–12 (2017)., Conti, J., Viola, M. G. & Camberg, J. L. *Mol. Microbiol.* **107**, 558–576 (2018)). Hence, a detailed characterization of the spatiotemporal regulation of FtsA by MinDE is not in the scope and also diverges too far off the main message of this manuscript: That the specific nature of the membrane protein to be regulated by MinDE *in vitro* is irrelevant. In this sense, FtsA is just another example for that the spatiotemporal regulation by MinDE is generic *in vitro*.

Thus, we have not further expanded on these experiments nor included them into the main text.

- In Supplementary figure 1d,e,f, the authors calculate and compare average MinD intensities. It is hard to follow the normalization that then authors did. Even though the difference between intensities in Min wave minima and maxima are very pronounced for these experiments, the denoted intensities don't seem to differ between panels d,e,f in the presented plots. It is therefore not clear how the authors obtained these data.

The average MinD intensities of the full image, in the MinDE minima and maxima have been obtained by segmenting the EGFP-MinD images into binary masks and subsequently multiplying those masks with the original images to obtain the pixels located in the MinDE minima or maxima (see Fig 1d). We then normalized all intensity values to the control condition, MinDE self-organization in the presence of His-mCh. In the previous version, we had accidentally written in the figure caption that fluorescence intensity was normalized to a fluorescent standard and then to the control case,

His-mCh. We have corrected this mistake that might have confused the reviewer. Of course EGFP-MinD fluorescence intensity in the MinDE minima and maxima differs, but because of the normalization they are all about 1. We have displayed the data like this to be comparable to the fluorescence intensities of the mCh-MTS constructs (Fig.3a-c) and to show that mCh-MTS addition has no influence on MinD densities on the membrane neither in total, nor in the maxima or minima of the MinDE wave.

- The Materials and methods section contains mistakes (e.g. in line 567 and 569, the authors use buffer at pH 80, numerical values are notoriously not spaced from units, which should be the standard according to SI Unit rules). This section should be carefully corrected.

We have carefully revised and corrected the methods section.

- The FtsA purification protocol (line 550) seems to be missing any lysogenic agent or procedure (e.g. lysozyme and sonication). The procedure is also missing protein purity control.

We have carefully revised and updated the FtsA purification and labeling protocol as we have substituted the experiments performed with maleimide-labeled FtsA with Sortase-labeled FtsA (new Supplementary Fig. 15).

- The manuscript is missing information regarding the temperature at which the experiments were performed.

We have added this information to the text in the section "Self-organization assays".

- Why did the authors not focus more on generic *E. coli* membrane proteins? I found it somewhat odd that in Fig.1 the authors study a *B. subtilis* protein as their prime/first example of a different protein that is modulated by the *E. coli* MinDE system.

We have used the mCherry fusion to the membrane targeting sequence from *B. subtilis* exactly because it is not an *E. coli* protein. While we do use several different *E. coli* peripheral membrane targeting sequences in Figure 2, namely MreB, FtsA and FtsY, we have used also the membrane targeting sequence of *B. subtilis* MinD to show that the spatiotemporal regulation is by no means a specific mechanism but a generic property of the system *in vitro*. We have added a sentence to the main text to clarify this point. We have also added a new Supplementary Fig. 7 showing that MinDE is also able to position mCherry proteins containing two copies of the native *E. coli* MinD MTS.

- The Materials and methods section lacks important information about the preparation of experiments with mCherry constructs and FtsA. For example, were these proteins added to the reaction volume or was the membrane first pre-incubated with these proteins and MinDE proteins were added later.

We have updated the materials and methods section with more information on the specific experiments.

- I dislike 'cheap sentences' like the last line of the abstract "This previously underestimated capacity of reaction diffusion systems to actively transport membrane proteins may be key to their function in bacteria and eukaryotes.". I suggest to be more modest and tone down the phrasing.

We have revised the manuscript including an updated discussion. However, we still like to adhere to the notion that this capacity of reaction-diffusion systems has been unexpected and thus, underestimated. How relevant it is to cells remains to be shown *in vivo*.

Reviewer #2 (Remarks to the Author):

This is an excellent study with novel and important discoveries. Previous studies from the Schwille lab have shown that waves of MinCDE on slb generate anticorrelated waves of FtsZ-mts. This was attributed to the effect of MinC in causing disassembly of FtsZ protofilaments and/or bundles. The present study shows that MinDE waves, without the inhibitor MinC, generate anticorrelated waves of non-specific proteins tethered to the membrane by mts amphipathic helices. A rather different wave is achieved by proteins linked by TM insertions, which can diffuse in the membrane but are not reversibly dissociating. This suggests that at least some of the inhibition of FtsZ by MinCDE may be due to steric inhibition on the membrane by MinDE, independent of MinC. Of course it would be interesting to see directly that MinDE can set up waves of FtsZ-mts and FtsZ-FtsA, but that can be a future study. The experiments presented here are convincing and well described. I recommend publication following attention to a few very minor concerns.

We thank the reviewer for his/her positive assessment of our work. We have added two new figures (new Fig. 5 and new Supplementary Fig. 16) with experiments showing the regulation of FtsZ-YFP-MTS. Please also see the answer to all reviewers above.

Biotinyl-CAP-PE needs a description. Fig. 3b shows two lipids inserted into the bilayer quite distant from each other. The Avanti web page shows a single lipid.

Biotinyl-CAP-PE is a single lipid. However, the streptavidin used in the experiment is a regular tetrameric protein. It has been shown that one tetrameric streptavidin is able to bind 2 or even 3 biotinylated lipids at the same time. (Dubacheva, G. V. *et al.*, *J. Am. Chem. Soc.* **139**, 4157–4167 (2017)). We have added a sentence to the main text and in the caption of Figure 3 to clarify this point.

Fig. 5d was not clear to me. In particular I did not understand “unit box” and the units -0.5 to +0.5.

We point the reader now to the Supplementary Fig. 14 (former 4) that describes the procedure and have added an explanatory sentence to the caption of Supplementary Fig. 14 (former Supplementary Fig.4) to better describe the analysis and resulting data. The MinDE oscillations vary within the microcompartments and hence result in slightly different positions of the maxima of the time-averaged concentration gradients along the long axis of the compartment. This is due to different protein amounts that are enclosed into the compartments when the buffer is lowered. To be able to compare different microcompartments, we determined the maxima of the EGFP-MinD gradients and projected those onto a unit box of length 1, from -0.5 to +0.5. We then also projected the time-averaged profile of the regulated component using the same coordinates.

Reviewer #3 (Remarks to the Author):

In this paper the authors further explore features of the Min system *in vitro* and show that peripheral membrane proteins counter oscillate with MinD. The pictures and movies are very convincing. This is shown in both their 2D and 3D systems and it is shown that it is fairly independent of the membrane targeting sequence they use. They also show that for a protein anchored to the membrane the behavior is quite different than that of the peripheral membrane protein. They also turn their attention to testing possible interaction of the Min system with DNA. This was stimulated by a report that MinD bound DNA. They find no evidence of MinD binding DNA but the Min pattern formation causes counteroscillations of the DNA in their system. The results lead to speculation about the possible influence of the Min system on proteins and DNA segregation that have not been directly linked to the Min system. In my opinion this is quite dangerous as the *in vitro* and *in vivo* systems are dramatically different as indicated below in my major concern.

We thank the reviewer for the positive assessment of our *in vitro* results and hope to address his/her point about the differences in the *in vitro* and *in vivo* system in the answer to all reviewers above.

Main concern.

The results are very clear in the *in vitro* system. My major concern is what it means. In the discussion the authors extrapolate to the *in vivo* situation. However, there is no data, only speculation. In this paper they estimate the MinD in the peak of the wave to be 16,000 per square micron. This is consistent with their *in vitro* explanation that MinD almost saturates the membrane and therefore excludes peripheral membrane proteins and acts as a diffusion barrier to proteins attached to the lipids. However, *in vivo* the estimate for MinD is about 200 per square micron (ref 12). Thus, the *in vivo* and *in vitro* systems are quite different. I don't understand why this wasn't addressed.

Please see the answer to all reviewers above.

The authors also looked at the potential interaction of MinD with DNA and found none. I do not find this surprising as the one report appeared badly flawed. The MinD mutant reported to affect MinD binding actually affect membrane binding.

Given that there is a number of research articles that show chromosome segregation defects *in vivo* when the MinCDE system is deleted that cannot be explained by mere cell division defects, the MinCDE system might actually be involved in chromosome segregation by a so far not determined mechanism (Jaffé, A. *et al.*, *J. Bacteriol.* **170**, 3094–3101 (1988); Mulder, E. *et al.*, *Mol. Genet. Genomics* **221**, 87–93 (1990); Åkerlund, T. *et al.*, *Mol. Microbiol.* **6**, 2073–2083 (1992); Jaffé, A. *et al.*, *J. Bacteriol.* **179**, 3494–3499 (1997); Åkerlund, T. *et al.*, *Microbiology* **148**, 3213–3222 (2002); Jia, S. *et al.*, *PLoS One* **9**, e103863 (2014)). A role for direct DNA binding to MinD for chromosome segregation was proposed by di Ventura *et al.*, supported by *in vivo* and *in vitro* data, as well as simulations (Di Ventura, B. *et al.* *Mol. Syst. Biol.* **9**, 686 (2013). As we could not detect direct binding of MinD to DNA in our *in vitro* assay, we considered that MinDE might be able to regulate DNA-membrane tethers. This is supported by the model in the report by di Ventura *et al.* that claims that a static or mobile gradient of DNA-membrane tethers is sufficient to aid chromosome segregation. We confirmed that MinDE are able to position DNA-membrane tethers in our setup *in vitro* irrespective of whether the DNA was tethered to the membrane *via* a cholesterol anchor or *via* biotin-streptavidin linkage. We do agree with the reviewer that this does not show that MinDE is participating in chromosome segregation using this mechanism *in vivo*. We have updated the discussion and removed the

schematic showing the model of how we envision chromosome segregation to be driven by MinDE oscillation to tone down our claims on such a mechanism. However, we feel it is important to show that we cannot detect direct MinD-DNA binding, whereas the spatiotemporal regulation of DNA membrane tethers by MinDE is strong *in vitro*.

In an earlier paper this group showed that Min counter oscillated with FtsZ. In that study the counteroscillation depended upon MinC; MinD/MinD alone could not do it. What is the difference with proteins here. Is it that FtsZ is polymerized?

We thank the reviewer for pointing out this discrepancy. Please see the answer to all reviewers above. We have added two new figures (new Fig.5 and new Supplementary Fig. 16) showing the regulation of FtsZ-YFP-MTS by MinDE.

Reviewer #4 (Remarks to the Author):

Ramm, et al., observed that the MinDE system can spatiotemporally regulate the distribution of model membrane-associated and membrane-anchored proteins *in vitro*. The authors propose that this is a novel/newly described feature of the MinDE system to systematically regulate the position of membrane-bound proteins by causing redistribution in the membrane. The authors also speculate that this mechanism may be important for regulating the midcell localization of cell division components, other membrane proteins and also modulating chromosome segregation *in vivo*. The data are presented in a very organized and well-communicated manner. The techniques and observations are interesting and compelling, but it is difficult to extend these observations to the *in vivo* physiological system. In an *in vivo* model, one could then test the relevant impact of tethering large complexes, or DNA, or other redistribution and compartmentalization scenarios including multispinning transmembrane domains.

We thank the reviewer for the positive assessment of our *in vitro* results. We agree with the reviewer that observing these effects *in vivo* would be interesting, but are postponed to future work as outlined in the answer to all reviewers.

There are several concerns regarding experimental constraints/parameters that impact the generalizable nature and relevance of the observations. The authors are asked to address the following comments.

1. Most of the experiments are performed at 1 μM MinDE in a chamber, where the surface waves of MinD represent regions of membrane-bound protein at high local concentration of MinD. The width of the MinD zone (i.e., in Fig. 1a) captured in the surface wave is 100 μm . The amount of MinD required to produce this zone seems disproportionately higher than the available MinD in a typical cell. If MinD is functionally capable of displacing surface associated proteins in an *in vivo* system, such as an *E. coli* cell, it must be densely populating the inner surface of the membrane. Is there enough MinD in a given cell to populate a membrane surface zone such that it behaves as in the system demonstrated *in vitro*. If there is, then is the zone expected to be smaller than the 100 μm zone depicted in the figure (1a)? And if substantially smaller, would it produce the same effects? What is the surface density required to displace? If the surface density of the min wave is increased, it should then be more productive at displacing/reorganizing. The cell-shaped microcompartments, if physiologically analogous or relevant, should also be designed to recapitulate MinDE cellular concentration, lipid content, membrane fluidity, etc.

For the first part of the comment please see the answer to all reviewers above.

The rod-shaped microcompartments are of course not an ideal mimic of the interior of a bacterial cell, but have been shown to recapitulate the pole-to-pole oscillations and protein gradient formation of MinCDE occurring *in vivo*. (Zieske, K. & Schwille, P. *Angew. Chemie Int. Ed.* 52, 459–462 (2013), Zieske, K. & Schwille, P. *Elife* 3, e03949 (2014)). Using this assay we can mimic MinDE pole-to-pole oscillations *in vitro* where we can precisely control all conditions without a complex cellular environment. Using the microcompartments we could show that MinDE are able to generate time-averaged protein gradients of functionally unrelated proteins that are maximal at mid-compartment. We have changed the sentence claiming that the microcompartments are a physiologically relevant condition, and just state that they mimic the pole-to-pole oscillations occurring *in vivo*.

2. The surface wave zones exhibit varying widths across the experiments. For example, zones depicted in Fig. 2b range from ~20 μm to ~60 μm . Are the zones traveling at different rates? What is the mean deviation of zone widths and deviation of the rate of the surface waves? What is responsible for this variation? Is this performed under steady-state conditions in which ATP is not limiting and ADP is not accumulating?

All our assays contain 2.5 mM ATP which is not limiting and allows the proteins to perform self-organization for more than 24 h. We have added a characterization of the wavelength and velocity for all mCh-MTS constructs including the control with His-mCh shown in Figure 2b to the Supplement (new Supplementary Fig. 4). While all the wavelengths and velocities have large errors, the mean and median values are very similar for all constructs and also in agreement with our previously published results (Kretschmer, S., Zieske, K. & Schwille, P. *PLoS One* 12, e0179582 (2017)). The wide spread of wavelength and velocities stems from the fact that even within the same assay chamber the wavelength varies, and also differs between experiments. Please see below the two tile scans of a sample chamber containing MTS(2xMreB-mCh) and MinDE. Independent of the defined wavelength or pattern generated by the MinDE system, MTS(2xMreB)-mCh generates patterns on the membrane that are a faithful negative image. The differences in the wavelength of MinDE patterns are probably caused by local differences in the supported lipid bilayer and the non-linear nature of the MinDE self-organization where small concentration differences between samples can lead to large differences in the patterns. The fluidity and properties of the supported lipid bilayer are also critical (resulting in either target patterns, spirals or parallel traveling waves), wavelength and velocity (Martos, A., Petrasek, Z., Schwille, P. *Environ Microbiol* 15, 3319-3326 (2013)). The membrane properties can change slightly from day to day due to differences in plasma cleaning of glass, difference in size of the small unilamellar vesicles used for SLB preparation, different lipid batches, air humidity and differences in washing of the SLB. Hence, to ensure comparability, all conditions in a subpanel in all Figures in the manuscript are always performed on the same day under the same conditions. The concentration of 1 μM MinD and 1 μM MinE used for the experiments here is especially variable as this concentration is at the border from the rather chaotic patterns formed for low MinE concentrations (1 μM MinD, <1 μM MinE) and more regular spiral and parallel traveling waves (1 μM MinD, >1 μM MinE) (see also Kretschmer, S., Zieske, K. & Schwille, P. *PLoS One* 12, e0179582 (2017), Loose, M., Fischer-Friedrich, E., Ries, J., Kruse, K. & Schwille, P. *Science* 320, 789–792 (2008), and the new Supplementary Fig. 5).

Independent of the exact wavelength or wave pattern generated MinDE are able to faithfully regulate mCh-MTS constructs as we show in our new Supplementary Fig. 5.

3. In addition to the strength of the interaction between the membrane-targeting region of the displaced protein and the membrane, a major impacting factor on redistribution of displaced protein should be membrane fluidity. If lateral diffusion in the membrane is affected, displacement, particularly by a permanently attached membrane protein, should also be affected. What are the effects of modifying membrane fluidity, either by altering lipid content or temperature, on redistribution of displaced protein. The permanent-attachment mimic used a nonpolar/lipid anchoring group that was fused to a protein. However, this would likely be far more diffusible than a full transmembrane domain. If diffusion is limited by reduced fluidity or a bulkier transmembrane(s), would the hindrance be sufficient to induce breakdown of the MinD surface wave propagation?

We use supported lipid bilayers formed on glass in our study, in which full-length transmembrane proteins are immobile as they come in contact with the support. We have used full transmembrane ZipA reconstituted in such supported lipid bilayer in the past and could not detect any lateral mobility of the protein and thus also no spatiotemporal regulation by MinDE (Martos, A. *et al. Biophys. J.* **108**, 2371–2383 (2015)). Hence, to observe the positioning of transmembrane proteins by MinDE, both proteins would need to be co-reconstituted on either free-standing membranes or cushioned supported lipid bilayers. Regrettably, this definitely very interesting experiment would go beyond the scope of this work and thus is postponed to future works.

We agree with the reviewer that streptavidin anchored to two to three biotinylated lipids will probably be diffusing significantly faster than a transmembrane protein if they were both to be reconstituted in a free-standing or cushioned model membrane. However, this might neither be a faithful image of diffusion on the inner membrane within a bacterial cell. For eukaryotic cells it has been shown that diffusion of membrane proteins is 5 to 50 time slower in the cell compared to free-standing membranes (Kusumi, A. *et al. Annu. Rev. Biophys. Biomol. Struct.* **34**, 351–378 (2005)). This could well be the case for bacterial membranes, as we have shown that the MinDE wavelength increases from in the cell, over reconstitution on SLBs to reconstitution on free-standing GUVs (cell: 3-8 μm , SLBs: 65-110 μm , GUV: 120-420 μm) (Martos, A., Petrasek, Z., Schwille, P. *Environ Microbiol* **15**, 3319-3326 (2013)).

Nevertheless, we can already draw important information about the impact on diffusion from the experiments contained in the manuscript.

When lateral diffusion of proteins that cannot detach from the membrane is entirely abolished, e.g. the streptavidin crystals that form at high streptavidin concentrations on the membrane (see Supplementary Movie 14) or the thick FtsZ-YFP-MTS bundles at high free Mg^{2+} concentration (new Fig.5, new Supplementary Fig. 16), these proteins cannot be laterally moved on the membrane by MinDE. These structures are also not able to “break down” the MinDE waves, instead MinDE waves couple across these obstacles similar to the coupling that has been shown to occur over membrane gaps (Schweizer, J. *et al. Proc. Natl. Acad. Sci.* **109**, 15283–15288 (2012)). In the case of FtsZ-YFP-MTS the strong bundles decrease the MinD density on the membrane, but MinDE still self-organize and regulate the FtsZ-YFP-MTS membrane attachment, albeit very weakly (new Fig. 5c-e).

Therefore we can conclude that a protein needs to be either able to attach or detach to and from the membrane or laterally diffuse in the membrane in order to be regulated by MinDE. We have added this important insight to the discussion, which also limits the amount of proteins that could potentially be regulated *in vivo*.

4. Does an mCherry with an MTS from *E. coli* MinD and/or MinE also become excluded? These fusions should have similar affinities for the membrane at MinD surface waves, excluding surface dimerization effects.

The *E. coli* MinD MTS by itself is too weak to support efficient binding of a protein to the membrane, but if two copies are supplied the membrane affinity strongly increases. This has been shown *in vivo* (Szeto, T. H. *et al., J. Biol. Chem.* **278**, 40050–40056 (2003)), where the MTS was fused to GFP as a single copy, as a tandem repeat or as a single copy to a dimerizing GFP leucine zipper fusion. Only the two latter constructs that supply two MTS allowed the GFP to efficiently bind to the membrane. This MinD MTS is also part of the chimeric FtsZ-YFP-MTS protein, which hardly binds to the membrane when no GTP is supplied, but strongly binds to the membrane when GTP is supplied and the protein can polymerize (Ramirez-Diaz, D. A. *et al. PLoS Biol.* **16**, e2004845. (2018)). This switch in membrane affinity from a single MinD MTS copy to two is also reflected in the MinDE system itself, as MinD needs to dimerize upon ATP binding to efficiently bind to the membrane.

To answer the reviewer’s question, we have constructed three new mCh-MTS constructs analogously to the proteins used by Szeto *et al.* and show the new results in Supplementary Fig. 7. One containing a single *E. coli* MinD MTS (mCh-MTS(1xMinD)), one containing a tandem repeat of the *E. coli* MinD MTS (mCh-MTS(2xMinD)) and a dimerizing construct containing mCherry fused to the leucine zipper Jun and the *E. coli* MTS (mCh-Jun-MTS(1xMinD)).

We have added a figure (new Supplementary Fig. 7) showing the behaviour of these proteins when co-reconstituted with MinDE. While mCh-MTS(1xMinD) membrane binding cannot be detected and is only very weakly regulated by MinDE, both constructs containing two copies of the MinD MTS, mCh-MTS(2xMinD) and the dimerizing mCh-Jun-MTS(1xMinD) bind to the membrane and are efficiently regulated by MinDE. This result is interesting in itself, as it questions the commonly accepted MinDE self-organization mechanism. If indeed the MinDE waves consisted of individual MinD dimers and no higher order organization or recruitment was present, these constructs should not be spatiotemporally regulated by the MinDE system. We have recently shown that MinDE seems to organize in transient higher order structures similar to a 2D crystal on the membrane during wave propagation (Miyagi, A., Ramm, B., Schwill, P. & Scheuring, S. *Nano Lett.* **18**, 288–296 (2017) and also other groups have reported higher order structures of MinD and MinCD (Suefuji, K., Valluzzi, R. & RayChaudhuri, D. *Proc. Natl. Acad. Sci. U. S. A.* **99**, 16776–16781 (2002), Conti, J., Viola, M. G. & Camberg, J. L. *FEBS Lett.* **589**, 201–206 (2015), Hu, Z., Gogol, E. P. & Lutkenhaus, J. *Proc. Natl. Acad. Sci. U. S. A.* **99**, 6761–6766 (2002). Hence, this finding supports the hypothesis that MinDE indeed pose a propagating diffusion barrier.

5. FtsA was suggested to behave as a surface associated protein (Fig. S5). However, FtsA has been reported to oligomerize on the membrane, promote membrane distortion, hydrolyze ATP rapidly and recruit FtsZ polymers to a membrane (Loose, 2014; Krupka, 2017; Conti, 2018). These activities likely complicate the predicted behavior and could explain what FtsA counteroscillations have not been observed *in vivo*.

We thank the reviewer for pointing out these important literature sources. Please see the answer to reviewer number 1 and our updated Supplementary Fig. 15 (former Supplementary Fig. 5).

6. The authors found that MinDE propagation on lipid bilayers can spatially regulate the membrane distribution of mCherry fusion proteins containing amphipathic helices and modeled transmembrane domains. Based on these observations, the authors propose a novel function of the MinDE system to spatiotemporally regulate membrane-associated and membrane-anchored proteins. Specifically, the authors propose that MinDE-driven counter oscillations of ZipA, which were previously reported (Bisicchia, 2013) leads to ZipA enrichment at midcell. By expressing Gfp-tagged ZipA in live cells, ZipA localization in wildtype cells can be monitored and compared to a minE deletion strain. If the MinDE system is important for septal localization of ZipA, then disrupting the MinDE system should lead to aberrant ZipA localization. If the ZipA transmembrane domain is replaced with an amphipathic helix, does the behavior or patterning change *in vivo*? This, and similarly designed experiments to test the predicted mechanism *in vivo*, as well as min-dependent localization of ZapB and MatP, which contacts the chromosome, would add physiological context to the study. In addition, distribution of fluorescent proteins *in vivo* in the min+ and min- strain could also be monitored for other proteins that are recruited to the membrane but do not participate in cell division, which would indicate the general applicability of the positioning or membrane-partitioning system.

We thank the reviewer for the detailed suggestions on experiments to perform *in vivo*. However, we believe that it is not in the scope of the manuscript to perform these *in vivo* experiments. Please see the answer to all reviewers above.

7. The authors observe that the spatial distribution of mCherry-MTS constructs is regulated by MinDE propagation *in vitro*. The authors confirmed that soluble His-mCherry is not spatially regulated by MinDE. However, if the concentration of His-mCherry were higher, would this remain true? Does molecular crowding affect the min surface wave or mass transfer?

We have shown in the past that the velocity and wavelength of MinDE patterns decreases in the presence of molecular crowders such as Ficoll400 (Schweizer, J. *et al. Proc. Natl. Acad. Sci.* **109**, 15283–15288 (2012)) or Ficoll70 (Martos, A. *et al. Biophys. J.* **108**, 2371–2383 (2015)), albeit never reaches the wavelength observed *in vivo*.

Crowding in solution should not have an influence on the spatiotemporal regulation observed herein, other than a decrease in wavelength and velocity of the MinDE patterns, because the mCh-MTS constructs and the lipid-anchored proteins are regulated on the membrane. We have performed a titration of MinDE concentration and of MTS(2xMreB)-mCh to show that the spatiotemporal regulation observed here is possible under a wide variety of MinDE concentrations in solution and hence membrane densities, and also occurs for different kind of MinDE patterns that are associated with different wavelengths (see new Supplementary Figures 5, 6). Furthermore, we show that the regulation is also occurring under reaction confinement in PDMS microcompartments (see Figure 4). The spatiotemporal regulation of the proteins observed is probably due to molecular crowding, not in solution, but on the membrane. The high density and possible higher order interaction between MinDE proteins on the membrane excludes other proteins from entering the dense MinDE wave either through attachment from solution (peripheral model membrane proteins) or through lateral

diffusion (lipid-anchored Streptavidin). Hence, the spatiotemporal regulation should be independent of the crowding in solution. Further, we show that MinDE self-organization is indeed influenced by crowding on the membrane. The lipid-anchored streptavidin densities used in our assay amount to a total surface coverage of about 17% and the MinDE membrane binding and wavelength is significantly affected by the presence of streptavidin on the membrane (Fig. 3d)

To show that even at high mCh-His concentration the protein is not regulated by MinDE, we have performed an experiment where we added 50 μM of His-mCh to the self-organization assay and cannot observe any spatiotemporal regulation of His-mCh even under these conditions (Please see images below).

8. Cells deleted for *minC* are associated with chromosome segregation defects *in vivo* (Akerland, et al., 2002), suggesting that MinC may contribute to the phenotype. It is possible that reduced *minDE* expression in *minC* mutants accounts for chromosome segregation defects, and that overexpression of *minDE* would promote proper chromosome segregation. A second possibility is that MinC alters surface wave propagation and reduces efficiency of directing chromosome segregation. The authors suggest that MinD could assemble into higher order structures (line 409). In fact, MinD is reported to form copolymers with MinC (Ghosal, et al., 2014; Conti, 2015).

We thank the reviewer for his/her comment on the importance of MinC in the system. In the previous version of the manuscript, we have shown that the spatiotemporal regulation of model peripheral membrane proteins and lipid-anchored proteins is occurring in the absence of MinC. However, MinC is an integral part of the MinCDE system *in vivo*. Hence, we have added a new figure (Supplementary Fig. 10) showing that MinDE also spatiotemporally regulates mCh-MTS constructs and lipid-anchored streptavidin when MinC is added to the assay.

We know from our previous *in vitro* experiments that MinC addition to the assay slightly alters MinDE waves by changing their velocity and wavelength (Loose, M., Fischer-Friedrich, E., Herold, C., Kruse, K. & Schwille, P. *Nat. Struct. Mol. Biol.* **18**, 577–583 (2011)). Further, we have shown that MinC levels of more than 1/5 of the amount of MinDE, (>0.2 μM MinC, 1 μM MinD, 1 μM MinE) disturb the MinDE wave propagation *in vitro* (Zieske, K. & Schwille, P. *Elife* **3**, e03949 (2014)). This finding is in agreement with the reported *in vivo* concentration of the MinCDE proteins of about 2000-3000 MinD, 1400 MinE and only about 400 MinC molecules per cell (de Boer, P. A., Crossley, R. E., Hand, A. R. & Rothfield, L. I. *EMBO J.* **10**, 4371–4380 (1991), Shih, Y. L., Fu, X., King, G. F., Le, T. & Rothfield, L. *EMBO J.* **21**, 3347–3357 (2002).) Hence, we agree with the reviewer that changes of MinC through deletion or mutation might have a direct effect on the properties of MinDE oscillations *in vivo* and that the situation *in vivo* is far more complicated and interdependent than in our *in vitro*

experiments. The strength of our experiments performed *in vitro* is that we can exactly control protein concentrations and omit any other factors that complicate the analysis.

9. The authors report that DNA bound to membrane tethers is spatially regulated by MinDE propagation on lipid bilayers independently of a direct protein-DNA interaction, and propose that the MinDE system can affect chromosome segregation *in vivo* by regulating the distribution of membrane-associated DNA binding proteins. FtsK is a membrane-anchored protein that regulates *E. coli* chromosome segregation and is localized to the septum (Yu, et al., 1998). If the MinDE system spatiotemporally regulates membrane-bound DNA binding proteins, then septal localization of FtsK-Gfp may be perturbed in *min*- cells. The authors are asked to examine the affect of disrupting MinDE on localization of membrane-anchored DNA-binding proteins in live cells.

We thank the reviewer for his detailed suggestion for *in vivo* experiments. Please see our answer to all reviewers above and the answer to comment 6.

REVIEWERS' COMMENTS:

Reviewer #1 (Remarks to the Author):

I think they have taken the criticism seriously and accommodated important changes. Hence I now support publication in Nature Comm

Reviewer #3 (Remarks to the Author):

The authors have responded to the previous criticisms with many new experiments and some modifications to the manuscript. There is no question that MinDE in the in vitro system drives a counter movement of a peripheral membrane protein. The major criticism in the previous reviews was the relevance to the in vivo situation. The authors responded with calculations and decreasing the MinD/MinE concentration about in half (from 1.0 to 0.4). However, the critical test would be to do an in vivo experiment rather than argue that they are avoiding the complexity of the in vivo system. They should have all the tools to do a simple test in vivo. They could use GFP-MinD/MinE in vivo with a min deletion strain and then express any one of their mcherry-MTS fusions that has good membrane affinity. They could then observe whether the counter oscillations between MinD and the mcherry-MTS take place in vivo.

Although the authors have toned down some of their claims I still think this needs to be improved. For example (pages 22-23) the authors state (line 483) "Several studies provided hints that MinDE oscillations influence chromosome segregation and the distribution and abundance of membrane proteins in vivo (sup Table 1). However, prior to the present study, this evidence had not been further corroborated." The authors are implying that their in vitro studies corroborate the in vivo hints. Corroborate means confirm and this is way too strong a statement.

Reviewer #4 (Remarks to the Author):

This manuscript reports that the Min system in bacteria is capable of promoting redistribution of membrane associated components, including proteins and DNA, in a controlled system in vitro and may further implicate this system in regulating localization of membrane associated components in vivo. Propagation of Min waves across a lateral surface could potentially serve to regulate the spatial distribution for a variety of physiological systems in vivo. This report presents important observations that will lead to additional investigations in vivo to test the relevance of the redistribution mechanism proposed.

REVIEWERS' COMMENTS:

Reviewer #1 (Remarks to the Author):

I think they have taken the criticism seriously and accommodated important changes. Hence I now support publication in Nature Comm

We thank the reviewer for the positive assessment of our revised manuscript.

Reviewer #3 (Remarks to the Author):

The authors have responded to the previous criticisms with many new experiments and some modifications to the manuscript. There is no question that MinDE in the in vitro system drives a counter movement of a peripheral membrane protein. The major criticism in the previous reviews was the relevance to the in vivo situation. The authors responded with calculations and decreasing the MinDMinE concentration about in half (from 1.0 to 0.4). However, the critical test would be to do an in vivo experiment rather than argue that they are avoiding the complexity of the in vivo system. They should have all the tools to do a simple test in vivo. They could use GFP-MinD/MinE in vivo with a min deletion strain and then express any one of their mcherry-MTS fusions that has good membrane affinity. They could then observe whether the counter oscillations between MinD and the mcherry-MTS take place in vivo.

Although the authors have toned down some of their claims I still think this needs to be improved. For example (pages 22-23) the authors state (line 483) "Several studies provided hints that MinDE oscillations influence chromosome segregation and the distribution and abundance of membrane proteins in vivo (sup Table 1). However, prior to the present study, this evidence had not been further corroborated." The authors are implying that their in vitro studies corroborate the in vivo hints. Corroborate means confirm and this is way too strong a statement.

We agree with the reviewer that to conclude that this non-specific regulation is occurring in vivo, detailed in vivo experiments are needed. We have further toned down our claims by removing the sentence in question.

However, we remain convinced that to prove that such a mechanism is occurring in vivo much more detailed studies than the simple test proposed are needed.

Our in vitro experiments in which we lowered the MinDE concentration to 0.4 μM resulted in membrane surface densities that are on the same order of magnitude as estimated densities in vivo. These experiments showed that while the spatiotemporal regulation is still visible, and importantly the overall mCh-MTS density on the membrane is reduced, it does get weaker with lower MinDE concentrations. Hence, expression levels of both MinDE and the target protein will potentially have an effect on the efficiency of the regulation in vivo. We cannot exclude that the regulation in vivo is rather weak and while the regulation might still have measurable effects, such as a reduction of overall membrane protein abundance or protein enrichment at midcell, the counter-oscillations might not be as clear and easily detected in vivo. Hence, to properly show the occurrence in vivo, it is necessary to work with fluorescent protein fusions that are expressed under the native promoter. Furthermore, much more advanced methods, such as proteomics or single-particle tracking, might be necessary. One of the likely targets ZipA has been shown to counter-oscillate to MinCDE in vivo, which has been explained by MinC depolymerizing FtsZ in a periodical fashion that in turn recruits ZipA. To disentangle if and to what extent MinDE might participate in this counter-oscillation by regulating ZipA, more sophisticated mutants are necessary. For instance, a simple test using a

ΔminCDE strain and expressing tagged MinD and MinE would not be possible, since it would alter the whole system, as FtsZ would not be depolymerized and hence would lock the anchors on the membrane.

However, as of today there are already clear hints from *in vivo* studies that the MinCDE system is involved in other processes in the cell apart from positioning MinC (see Supplementary Table 1), e.g. a proteomics study showed that the abundance of several peripheral membrane proteins is reduced by the presence of MinCDE (Lee, H.-L. *et al.* Quantitative proteomics analysis reveals the Min system of *Escherichia coli* modulates reversible protein association with the inner membrane. *Mol. Cell. Proteomics* **15**, 1572–1583 (2016)).

Hence, we postpone a detailed *in vivo* study to future works, but would like to point out that our results are interesting independent of their *in vivo* relevance. First, from a purely biophysical point of view: a propagating diffusion barrier of proteins that themselves do not move in a directed fashion, drives the directed movement of another protein. Second, the results obtained here will allow to use MinDE as a non-specific regulator of membrane-bound components in artificial cells.

Reviewer #4 (Remarks to the Author):

This manuscript reports that the Min system in bacteria is capable of promoting redistribution of membrane associated components, including proteins and DNA, in a controlled system *in vitro* and may further implicate this system in regulating localization of membrane associated components *in vivo*. Propagation of Min waves across a lateral surface could potentially serve to regulate the spatial distribution for a variety of of physiological systems *in vivo*. This report presents important observations will lead to additional investigations *in vivo* to test the relevance of the redistribution mechanism proposed.

We thank the reviewer for recognizing the importance of our work.